# descSPIM: an affordable and easy-to-build light-sheet microscope optimized for tissue clearing techniques

Kohei Otomo [1,2,3,4,5,30], Takaki Omura[1,3,6,7,30], Yuki Nozawa [2], Steven J. Edwards[8], Yukihiko Sato[1,3], Yuri Saito[1,3], Shigehiro Yagishita[9,10], Hitoshi Uchida [11], Yuki Watakabe [4,5], Kiyotada Naitou [12], Rin Yanai[13], Naruhiko Sahara[13], Satoshi Takagi [14], Ryohei Katayama [14], Yusuke Iwata [15], Toshiro Shiokawa[15], Yoku Hayakawa[15], Kensuke Otsuka[16], Haruko Watanabe-Takano [17], Yuka Haneda [17], Shigetomo Fukuhara [17], Miku Fujiwara[18], Takenobu Nii[18], Chikara Meno [18], Naoki Takeshita [19,20], Kenta Yashiro[19], Juan Marcelo Rosales Rocabado [21], Masaru Kaku [21], Tatsuya Yamada [22], Yumiko Oishi[23], Hiroyuki Koike [23], Yinglan Cheng[23], Keisuke Sekine [24], Jun-ichiro Koga [25], Kaori Sugiyama[26], Kenichi Kimura [27], Fuyuki Karube[28], Hyeree Kim [29], Ichiro Manabe [29], Tomomi Nemoto[4,5], Kazuki Tainaka [11], Akinobu Hamada[9,10], Hjalmar Brismar [8] & Etsuo A. Susaki [1,2,3] ✉

Despite widespread adoption of tissue clearing techniques in recent years, poor access to suitable light-sheet fluorescence microscopes remains a major obstacle for biomedical end-users. Here, we present descSPIM (*des*ktop-equipped SPIM for *c*leared specimens), a low-cost ($20,000–50,000), low-expertise (one-day installation by a non-expert), yet practical do-it-yourself light-sheet microscope as a solution for this bottleneck. Even the most fundamental configuration of descSPIM enables multi-color imaging of whole mouse brains and a cancer cell line-derived xenograft tumor mass for the visualization of neurocircuitry, assessment of drug distribution, and pathological examination by false-colored hematoxylin and eosin staining in a three-dimensional manner. Academically open-sourced (https://github.com/dbsb-juntendo/descSPIM), descSPIM allows routine three-dimensional imaging of cleared samples in minutes. Thus, the dissemination of descSPIM will accelerate biomedical discoveries driven by tissue clearing technologies.

Since the early attempts of organ-scale three-dimensional (3D) imaging with light-sheet fluorescence microscopy (LSFM)[1], tissue clearing has become the gold standard for volumetric tissue, organ, and body imaging. Highly effective tissue clearing techniques for LSFM imaging include benzoic acid benzyl benzoate (BABB) based clearing[2], ethyl cinnamate (ECi) based clearing[3], 3D imaging of solvent cleared organs (3DISCO)[4], clear, unobstructed brain/body imaging cocktails and computational analysis (CUBIC)[5,6], *m*-xylylenediamine (MXDA)-based aqueous clearing system (MACS)[7], polyethylene glycol (PEG) associated solvent system (PEGASOS)[8], small micelle improved human organ antibody efficient labeling (SHANEL)[9], cleared lipid extracted acryl hybridized rigid immunostaining/in situ hybridization compatible tissue hydrogel (CLARITY)[10], and stabilization under harsh conditions via intramolecular epoxide linkages to prevent degradation

(SHIELD)[11]. The maturation of tissue clearing protocols and their subsequent commercialization has enabled adoption by end-users and supported numerous scientific discoveries across a wide range of biomedical research fields[12–15].

Two prominent variants of LSFM, selective plane illumination microscopy (SPIM) and digital scanning light-sheet microscopy (DSLM), were proposed prior to the extensive development of tissue clearing techniques in 2010's[16,17]. Their original application was to image small and intrinsically translucent specimens such as fly embryo and fish larva in 3D and 4D manner[16,17]. SPIM utilizes a cylindrical lens to generate a static excitation light sheet. DSLM generates a light-sheet by scanning a narrow-focused beam in one axis using a galvanometer mirror. The excitation light sheets utilized in both SPIM and DSLM are generated by focusing a Gaussian beam, causing a tradeoff between axial spatial resolution and effective field of view (eFOV)[18]. To overcome this, tilling LSFM[19] and axial sweeping schematics[20] were proposed and successfully enlarged the eFOV with maintaining axial spatial resolution. The original DSLM approach has been extended to improve performance whilst increasing optical complexity[18]. One of the most prominent examples is the contrast enhancement achieved by synchronizing the galvanometer mirror with the rolling shutter of the camera[21,22], which has a similar effect to confocal slit detection[23]. Moreover, recent state-of-the-art approaches utilizing non-diffracting light sheets such as Bessel beam[24–26], Airy beam[27] and optical lattice-based light sheets[28] are reported based on modifications of DSLM.

Most LSFMs have adopted an orthogonal arrangement between the excitation and detection objectives, introducing physical restrictions to the size and mounting of the specimens. Conventional sample holders, such as culture dishes and well plates could not be easily accommodated in the microscope. Inverted geometry systems, such as dual-view inverted SPIM (diSPIM)[29,30] and open-top light-sheet (OTLS) microscopy[31–33], were later developed to provide ease of use in measurements of larger samples. Moreover, single-objective lens LSFMs, such as swept confocally-aligned planar excitation (SCAPE)[34,35] and single-objective SPIM (soSPIM)[36] have also been developed for achieving high-spatial-resolution imaging.

LSFM has also been optimized for rapid, volumetric-cleared tissue imaging. As the first proof of concepts, Dodt et al. developed a macro zoom microscope-based SPIM equipped with a low numerical aperture (NA) objective lens[1]. Such low NA systems have been improved in both image quality and simplicity by adding various options adapted to the specimen[37–40]. On the other hand, several advanced ideas are being proposed to achieve detection with a higher NA objective lens, improved axial resolution, and wider field of view (FOV). These systems include CLARITY-optimized light-sheet microscopy (COLM)[41], tilling LSFM[42], axial sweep schematics[43], and confocal slit detection[23]. For improving imaging speed with a high NA detection system, a moving observation with an efficient real-time autofocus (MOVIE) system was also proposed[44].

However, the limited accessibility of LSFM systems remains a major obstacle for many tissue-clearing end-users (Fig. 1a). Commercialized systems provided ease of use, while being prohibitively costly (>$500k) for individual research groups. Conversely, cutting-edge custom-built microscopes, plotted as the "advanced system" in Fig. 1a, require a high-level of expertise in optics to construct. mesoSPIM and Benchtop-mesoSPIM are successful open-source projects that reduced the barrier to some degree[38,40]. Nevertheless, the system still targets imaging core facilities run by microscopy experts overseeing its installation and maintenance (Fig. 1a and Supplementary Fig. 1). Other open-source projects in the early days, such as openSPIM[45] and OpenSpin microscopy[46], are optimized for live imaging of small model organisms but are not suitable for specimens treated with modern tissue clearing methods (Supplementary Fig. 1). More easy-to-build do-it-yourself microscopy including LEGO®-based LEGOlish (https://www.microlist.org/listing/legolish/), smartphone-

based miniSPIM[47], 3D-printer-based UC2 (You. See. Too.)[48], and pocket laser projectors-based projected light sheet microscopy (pLSM)[49] utilizes modern and cost-effective technologies. To obtain a significant scientific performance, however, the exclusion of specialized optical components may be controversial.

Here, we present descSPIM, a simple, low-cost, yet practical system designed specifically for tissue-clearing technique users in biomedical fields unfamiliar with microscopy systems. All components, with the exception of laser light sources, can be purchased from a single microscopy parts supplier. The total price, including laser light sources, ranges from $20,000 to $50,000. As part of our open-source initiative, we provide a parts list as well as construction and usage instructions. descSPIM enables efficient 3D imaging at $3.45 \times 3.45 \times 7$–$25\ \mu m^3$ voxel resolutions despite its basic SPIM configuration. We have demonstrated rapid 3D imaging of a cleared mouse brain hemisphere and a 2-mm thick mouse brain section, as well as advanced imaging of the entire mouse brain and xenograft cancer tissue. This user-friendly, and cost-effective LSFM system thus enables routine analysis of cleared samples at the researchers' workbench.

## Results
### Configuration of descSPIM
The invention of descSPIM necessitated a fundamental redesign of all configurations for functionality and minimization. The system basically adopted the original SPIM configuration for illumination optics[16] instead of DSLM[17], which requires additional mechanical controls of galvanometer mirrors. All the components for the excitation/detection optics and stages (approximately 90 parts) were aligned on a 300 mm × 450 mm breadboard, forming a one-sided light-sheet illumination with minimal equipment (Fig. 1b–d and Supplementary Fig. 2). They can be purchased from a single vendor (Thorlabs) for the user's convenience, in contrast to other open-source projects using several custom parts (Supplementary Table 1 and Supplementary Fig. 1).

A single-mode fiber-guided laser light was collimated with an achromatic convex lens, then formed into sheet illumination by a cylindrical lens. Users can choose color variations according to their experimental purpose. We used a four-color laser light source (Cobolt Skyra 488/515/561/647 nm) for multichannel imaging.

The sample stage included manual $x$, $y$ and $\theta$ stages to reduce costs, and a motorized $z$ stage for imaging. In addition, another motorized $z$ stage was attached to the detection optics for focusing. The cleared sample was placed in a four-sided transmission cuvette so that the users could handle it without an imaging oil chamber (Fig. 1c, d and Supplementary Fig. 2). One of the $z$ stages moved the sample, while the other shifted the detection optics. The latter stage was required to get into a focused position and to prevent defocusing during imaging, which was caused by the change in the distance ratio between the clearing reagent with a high refractive index (RI) and the air with a RI of 1.00 (Supplementary Fig. 3).

The detection optics included an objective lens with a low numerical aperture (NA) and long working distance (WD) (Thorlabs TL2X-SAP) combined with a tube lens and a CMOS camera with a rectangular sensor (size 14.131 × 7.452 mm²). The sensor shape and size allowed the effective width of the light-sheet to cover the whole sensor area as a FOV (Fig. 1e and Supplementary Fig. 4a). The resulting images, acquired at an effective integration magnification of 1×, had a pixel size of $3.45 \times 3.45\ \mu m^2$.

For the practical use of the basic configuration, two types of cylindrical lenses were adopted to create wider or axially finer illumination (Fig. 1e and Supplementary Fig. 4a). Full-FOV (FF) illumination was generated by one of the cylindrical lenses ($f = 500$ mm). The other cylindrical lens with a shorter focal length ($f = 150$ mm) yielded fine-axial (FA) illumination. We estimated their lateral and axial resolutions by full-width of half maximum (FWHM) values of point spread function

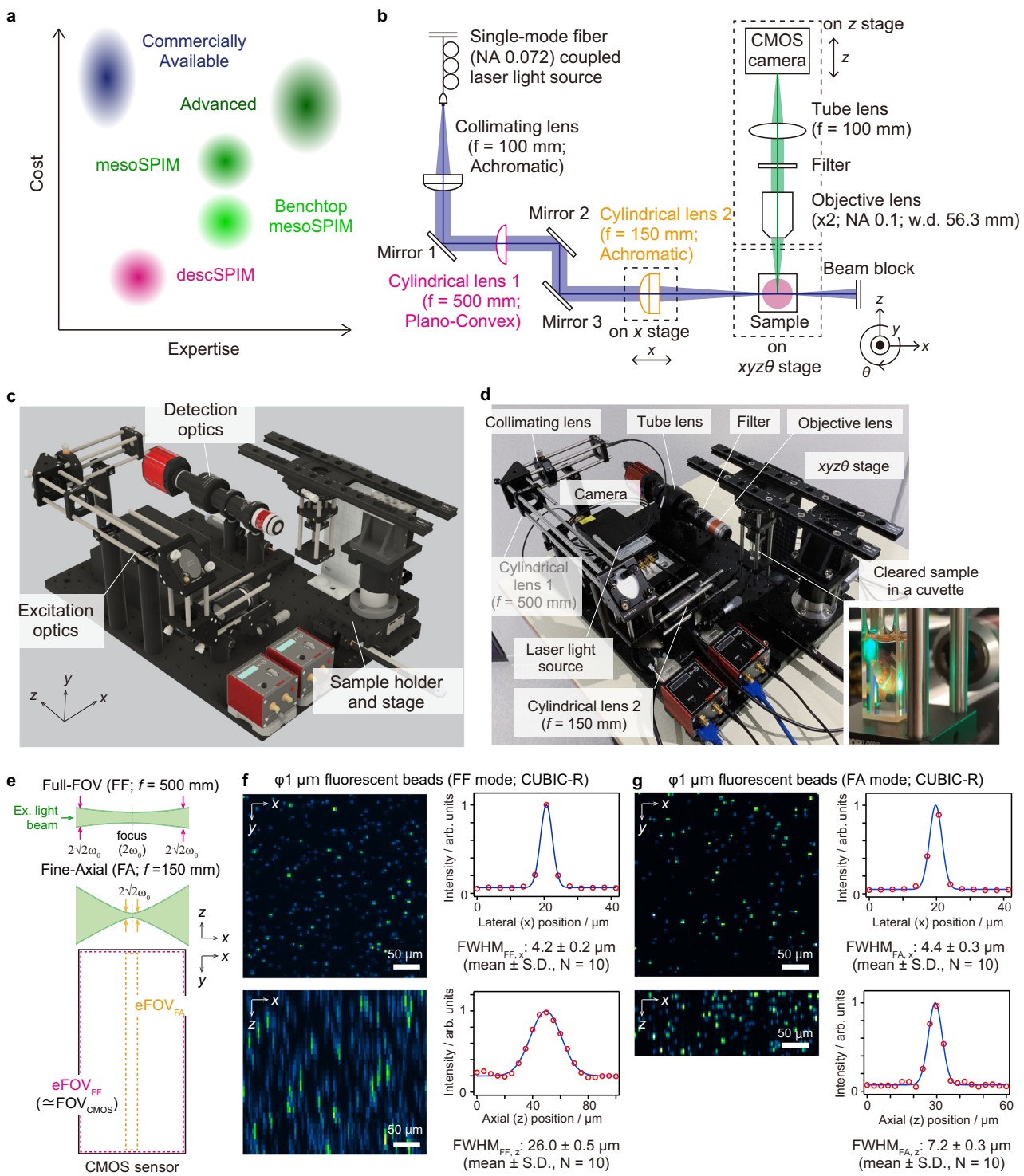

**Fig. 1 | descSPIM concept, components, and optical specifications. a** The target of descSPIM on the expertise and cost axes, where the simultaneous achievement of thorough simplification and practical quality is required. **b** descSPIM system overview. A one-sided light-sheet illumination formed by a collimated lens and a cylindrical lens comes from the right side of the sample. Either a cylindrical lens ($f = 500$ mm or $f = 150$ mm) is used for each illumination mode (Full FOV; FF and Fine axial; FA), respectively. A manual linear translation stage is attached to the latter cylindrical lens for the tilling light-sheet (TLS) imaging. The detection path is composed of a 2× objective lens, tube lens, and CMOS camera, with an effective integration magnification of 1× (3.45 μm × 3.45 μm of the pixel size). **c** Schematic overview of the descSPIM. **d** Overview of the actual descSPIM instrument. (inset) The cleared sample can be placed in a four-sided transmission cuvette and directly illuminated for imaging. No oil chamber was adopted. **e** Two illumination modes adopted to the descSPIM by switching the cylindrical lenses. FF covers the whole CMOS sensor area (7.45 mm in width). FA provides finer axial resolution while covering ~11% (830 μm of the Rayleigh length in width) of the sensor area. eFOV: effective field of view. **f, g** Estimation of lateral and axial resolutions by point spread functions (PSFs; evaluated with images 1 μm beads images). Comparable $xy$ resolution was achieved by the two illumination modes. FA provides approximately 7 μm of axial full width of half maximum (FWHM) value, over three-times finer than FF mode. The original grayscale 8-bit maps were pseudo-colored with a Green Fire Blue look-up table. The imaging experiments were performed twice with nearly identical results.

(PSF) that was defined from the intensity profile of the signals from a Φ1 μm bead embedded in CUBIC-R-agarose (Fig. 1f, g). Their lateral (x-y) resolutions were comparable (4.2 ± 0.2 μm for FF mode and 4.4 ± 0.3 μm for FA, respectively). The axial resolutions (x-z or y-z) of FF and FA modes were 26.0 ± 0.5 μm and 7.2 ± 0.3 μm, respectively, consistent with our design intent regarding the light-sheet shapes. Indeed, the effective field of view (eFOV, defined as 2 × Rayleigh length[18]) (Supplementary Fig. 4b) of FF mode largely covered the CMOS sensor area (>7.452 mm) (Supplementary Fig. 4c) instead of the lower axial resolution. On the other hand, FA mode provided a restricted width of eFOV (Supplementary Fig. 4d). These estimated light-sheet shapes were further confirmed by the direct measurement of the beam profiles in the air (FF mode: axial FWHM, 25.1 ± 0.0 μm, eFOV, 11.0 mm; FA mode: axial FWHM, 7.9 ± 0.3 μm, eFOV, 889 μm) (Supplementary Fig. 5). We considered practical z-step ranges to be roughly half of FWHM, with a minimum of 10 or 5 μm for FF and FA modes, respectively, for the cleared tissue imaging in this paper.

The FWHM values were also utilized to extrapolate the literature-defined effective NA value of cylindrical lenses for SPIM[18] (Supplementary Fig. 4b). We determined the effective NA values to be 0.007 for the FF mode and 0.033 for the FA mode. Based on the NA value, the eFOV in FA mode is approximated as 830 μm (Supplementary Fig. 4b), which is consistent with the eFOV estimation from an image of Φ1 μm beads (Supplementary Fig. 4d). However, the 830 μm measurement is narrower than the estimated effective field of view (eFOV) in the clearing reagent based on the measured beam profiles in the air (Supplementary Fig. 5). The higher RI elongates the Rayleigh length by a factor equal to the RI value compared to its length in the air. The pixel size of our utilized beam profiler was 4.4 × 4.4 μm², which is comparable to the estimated axial FWHM of the light intensity profile. This similarity might result in an overestimation of the light intensity profile. The eFOV calculated in FF mode at NA 0.007 is 16 mm in an environment with a refractive index of 1.52, which aligns with the measured value of 11 mm in the air (Supplementary Fig. 5). With these considerations, the cylindrical lens used in FA mode yields a sheet illumination near to the approximation. Since the ratio of NA values is equalized to the ratio of focal length, the effective NA of the cylindrical lens used in FF mode based on the NA of the cylindrical lens in FA mode (0.033) should be 0.01 (0.033 × 150 mm/500 mm), which is inconsistent with the effective NA of the cylindrical lens in FF mode (0.007). This discrepancy could be attributable to the aberration in the plano-convex lens used in FF mode, which led to a lower effective NA. Consequently, the cylindrical lens in FF mode generated a homogeneous light sheet illumination (approximately 26–29 μm of FWHMs) across the CMOS sensor area (Supplementary Fig. 4c).

Data acquisition and processing procedures are summarized in Supplementary Fig. 6. The basic system does not necessarily require users to deploy any custom programs for operations; it can be used only with the device-associated software. Two actuators should be moving synchronously at distinct velocities. The synchronous speed correction value (the relative velocity of two actuators) can be determined by the method in Supplementary Fig. 6. Inspired by the MOVIE method[44], a z-stack image was rapidly collected as time-lapse (xy-t) data by the continuous moving of the actuators, and then the data was converted into xy-z format. This method enabled rapid volumetric imaging, typically finishing an acquisition of a single z-stack in minutes.

## Imaging performance of descSPIM

As representatives, we collected a CUBIC-cleared and propidium iodide (PI)-stained mouse hemisphere and a 2 mm-thick mouse brain slice acquired with FF and FA modes, respectively (Fig. 2a–d, Supplementary Movie 1 and Supplementary Movie 2). The speed was 3 – 6 minutes/stack, depending on the illumination mode. To homogenize the illumination intensity along the y-axis, an image of dye solution (1 μM fluorescein in CUBIC-R) was parallelly obtained for

reference in order to adopt a flat-field correction (FFC) of Gaussian illumination[50] (Supplementary Fig. 7a, b). We further installed a manually moving stage for the cylindrical lens (Supplementary Fig. 8a), taking up the tiling light-sheet method (TLS)[19] into the FA mode imaging (Supplementary Fig. 8b). The TLS successfully achieved a uniform axial resolution across the sample width on the x-axis (Supplementary Fig. 4c). The resulting axial resolutions for the nuclei, estimated by FWHM of PSF, were consistent with the results by Φ1 μm beads (Figs. 1f, g, 2b, d). In the image obtained by FF mode, the axial FWHMs at the center and edge positions of the FOV were comparable (29.5 ± 0.2 μm and 27.0 ± 0.1 μm), again suggesting the homogeneous light-sheet thickness across the eFOV (Fig. 2a, b). As for the image with FA mode, the elongation of axial FWHMs compared with lateral FWHMs (11.4 ± 0.7 μm in axial and 8.2 ± 0.8 μm in lateral, approximately 1.37× elongation) was reduced compared with the axial elongation obtained with Φ1 μm beads (7.2 ± 0.3 μm in axial and 4.4 ± 0.3 μm in lateral, approximately 1.64× elongation) (Figs. 1g, 2d), indicating that the FA mode provides a quality of image close to the isotropic resolution for the nuclei-size objects.

We validated the reproducibility of z-stack measurements and identified axial shifts in the stack resulting from disparities in start timing (Supplementary Fig. 9a). However, these discrepancies were effectively mitigated through adjustments solely in the z-direction (Supplementary Fig. 9b). Notably, no shifts were detected in the xy-plane, ensuring consistent acquisition reproducibility (Supplementary Fig. 9a). To address axial displacements, we provided custom software allowing users to specify start timing and ensure z-position (Supplementary Fig. 10). Similar to Supplementary Fig. 9, this acquisition method was verified to enable reproducible image acquisition (Supplementary Fig. 11). Indeed, it effectively reduced axial displacement, rendering it an ideal option for experiments requiring precise position reproducibility (Supplementary Fig. 11).

We also tested the compatibility of descSPIM with widely used organic solvent-based clearing reagents, BABB[2] and ECi[3]. Occasionally, solvent-based clearing protocols necessitate special consideration for the reagent evaporation and resistance of microscopy equipment to the reagent. However, due to the use of a compact glass cuvette as the sample chamber, such considerations may not be taken into account when imaging with descSPIM. Protocol-specific differences in the refractive indices of RI-matching reagents should also be considered. Due to its synchronized two actuators moving with a relative velocity value specific to the sample and reagent RI, descSPIM can also circumvent this issue. To evaluate PSFs in BABB[2] and ECi[3], we estimated their lateral and axial resolutions in FA mode by FWHM values of the intensity profile of the signals from a Φ15 nm quantum dots embedded in gelatin gel substituted with organic solvents (Supplementary Fig. 12a). The lateral (x-y) resolutions were measured as 4.8 ± 0.3 μm (N = 10) in BABB and 4.6 ± 0.4 μm (N = 10) in ECi, showing negligible deviation from the values obtained in CUBIC-R-agarose (Fig. 1g). However, the axial resolutions (x-z or y-z) were found to be 9.5 ± 0.7 μm (N = 10) in BABB and 9.8 ± 0.3 μm (N = 10) in ECi, representing a degradation of over 30% compared to the values in CUBIC-R-agarose (Fig. 1g). This degradation stemmed not only from the longer excitation wavelength of approximately 50 nm but also from spherical aberration resulting from the RI difference between the organic solvent and the glass cuvette (RI = 1.56 for BABB and ECi and 1.52 for BK7 glass). By extrapolation of FWHM values to the literature-defined effective NA value of cylindrical lenses for SPIM[18] (Supplementary Fig. 4b), we determined the effective NA values for the FA mode to be 0.026 in BABB and 0.025 in ECi. Utilizing these NA values, the eFOV in FA mode was estimated to be approximately 1488 μm in BABB and 1600 μm in ECi. To demonstrate 3D biological imaging, we obtained a BABB-cleared hemisphere of the mouse brain that was stained with TO-PRO®−3 nuclear stain dye and perfused Evans blue dye in FF mode (Supplementary Fig. 12b). The imaging results indicated that

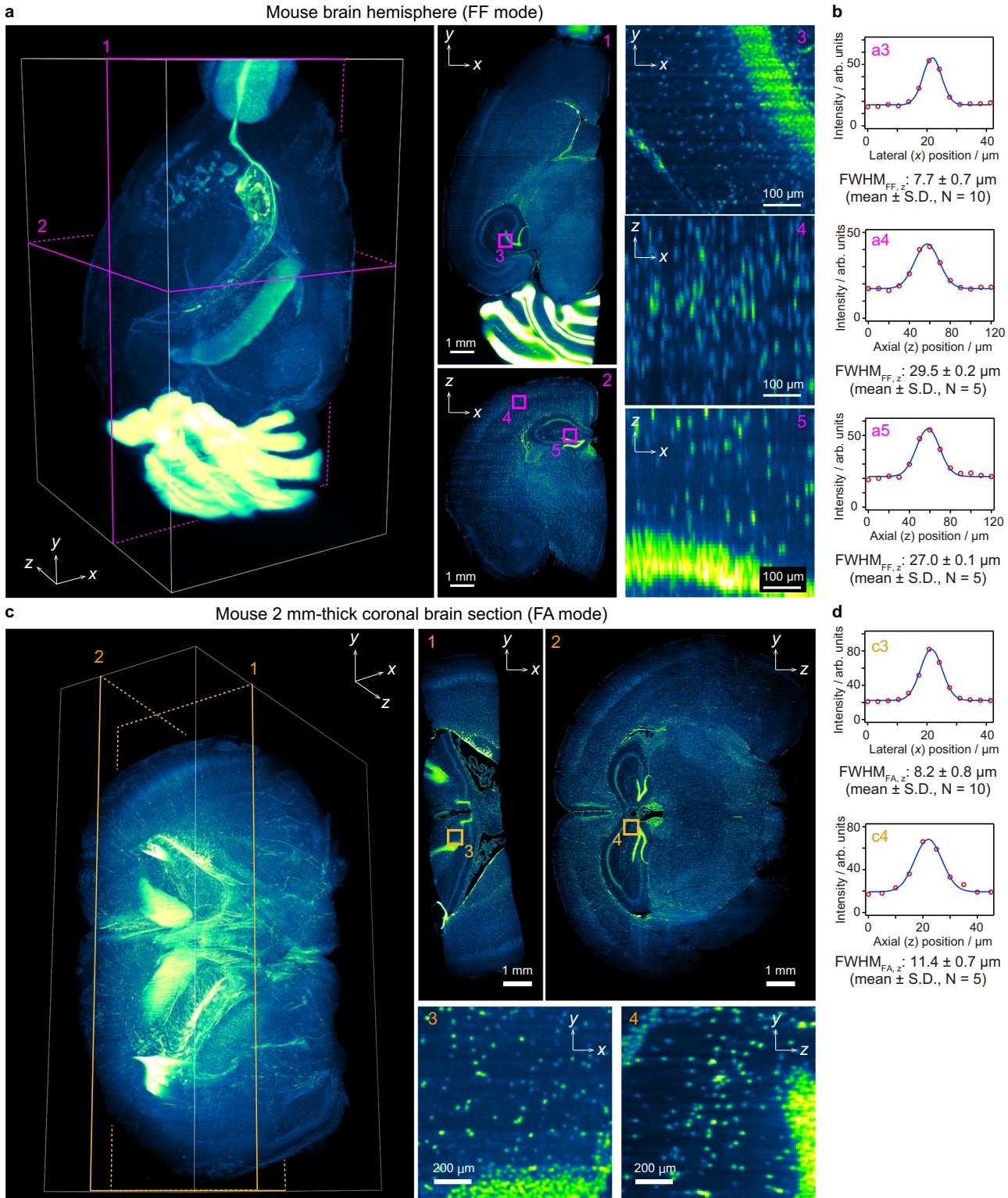

descSPIM effectively provided 3D imaging data for the BABB-cleared sample with a different refractive index than CUBIC-R+ (RI = 1.52). Thus, 3D imaging with descSPIM gives robust results independent of tissue clearing protocols.

### Advanced imaging and data processing for neuroscience application

Modern tissue clearing and 3D imaging techniques have been in high demand, particularly in neuroscience. To demonstrate the practical neuroscience applications of descSPIM, we acquired a volumetric whole-brain image of a PI-stained *Thy1*-YFP-H Transgenic (Tg) mouse[51]. To compensate for the restricted FOV size and optical attenuation along the direction of light-sheet illumination, we obtained multi-positioned, multi-directional stacks in FF mode (Supplementary Fig. 13a–c). In general, image processing poses a significant challenge for many end users in practical applications. To alleviate this burden, we have developed semi-automated custom codes tailored for registration and merging tasks. The corresponding stacks from two

**Fig. 2 | Optical specifications and imaging results. a** Full FOV (FF)-mode imaging of a propidium iodide (PI)-stained mouse brain hemisphere. Voxel size: 3.45 × 3.45 × 10 μm³. Imaging speed: 180 s. per 900 slices. Data size: 8.02 GB per 8-bit image stack. Flat-field correction (FFC) was applied. The *xy* images illustrate a homogeneous image contrast across the FOV. The *yz* images indicate the elongation of PI-stained cell nuclei across the *z* direction due to the lower axial resolution. Note that the basic system did not include a device for eliminating the stripe shadow artifact (e.g., a galvanometer mirror) for simplification. Despite these technical limitations, the reconstituted 3D image is qualitatively sufficient for evaluating the whole sample structure. The original grayscale 8-bit maps were pseudo-colored with a Green Fire Blue look-up table (LUT). The imaging experiment was performed at least 5 times with nearly identical results. **b** Lateral and axial spatial resolutions of PI-stained nuclei obtained with FF mode, estimated by the full width of half maximum (FWHM) values of their intensity profiles. The axial FWHM values were consistent with the value measured by beads (Fig. 1f) and about quadrupled by the lateral FWHM of the PI signals. **c** Fine axial (FA) mode imaging of a PI-stained, 2 mm-thick mouse brain section. Voxel size: 3.45 × 3.45 × 5 μm³. Imaging speed: 372 s per 1860 slices per stack. Data size: 16.54 GB per 8-bit image stack. A total of four stacks with different light-sheet focus positions were collected, then tilling light-sheet (TLS) was applied. The image intensity was corrected by FFC. The original grayscale 8-bit maps were pseudo-colored with a Green Fire Blue LUT. The imaging experiment was performed twice with nearly identical results. **d** Lateral and axial spatial resolutions of PI-stained nuclei obtained by FA mode, estimated as in (**b**). The *yz* images indicate that the FA mode provides a quality of image close to the isotropic resolution for the object with cell nuclei size across the *z* direction.

orientations (0° and 180° angles) were then registered using ANTs[52], followed by fusion into a single complemented stack (Supplementary Fig. 13d). To this end, we compared registration accuracy when obtaining transformation matrices using various degrees of downsized stacks (50% or 25%) (Supplementary Fig. 14a). The brain-wide registrations seemed to be completed in both the 25% and 50% downsized (Supplementary Fig. 14b). When examining the images in pixel order, however, misaligned signals were observed when using 25% compressed data: nuclei or neuronal somas were occasionally duplicated in the registered image (Supplementary Fig. 14c and d). Using 50% downsized data improved the registration accuracy (Supplementary Fig. 14c, d). Therefore, we processed the images with 50% downsized data. A high-end workstation (see Methods) was required to calculate the transformation matrices. Using the constructed data processing procedure (Supplementary Fig. 14d), we successfully registered and fused the two directional stacks in each channel (Supplementary Fig. 13e). Finally, we stitched all the fused stacks from distinct FOVs using BigStitcher[53] to reconstruct a single, complemented whole-brain stack (Fig. 3a, Supplementary Figs. 13b, 15, and Supplementary Movie 3). The resulting two-color image of the entire brain revealed the distributions of cell bodies and neurites, indicating that descSPIM provided 3D neural images with cellular resolution or more (Fig. 3b). For further comparison, we acquired a 3D image of another *Thy1*-YFP-H brain with our advanced light-sheet microscopy system (GEMINI[39]). Comparing the FWHM of the axial intensity profile of somas imaged by descSPIM (Fig. 3b) and GEMINI (Supplementary Fig. 16), we found that the axial resolution of descSPIM in FF mode was 32.0 ± 1.1 μm ($N = 20$), while the GEMINI's resolution was 17.5 ± 1.6 μm ($N = 20$). Therefore, although descSPIM can support the 3D imaging applications on an organ scale required for neuroscience research, its axial resolution did not reach a similar level when compared with an advanced system. Switching to high-resolution FA mode would be necessary for some applications requiring a higher axial resolution. Moreover, since the final data size is approximately the same for both the GEMINI and multi-view datasets, we believe that handling data from multiple stacks obtained through this acquisition method would not pose a significant burden on modern 3D imaging systems that handle recent data.

We performed an application in which two color channels (nuclear staining and target signal channels) were acquired with a single laser light excitation to simultaneously demonstrate the concept of descSPIM as a low-cost microscope and its capability for whole-brain imaging. With a single excitation wavelength (488 nm), we again obtained multi-positional stacks of the PI-stained *Thy1*-GFP-M Tg mouse brain. The stacks were effectively stitched with BigStitcher and reconstituted as a stack covering the entire brain area. Using appropriate bandpass filters, the EGFP and PI signals excited by a single-wavelength laser light were successfully separated (Supplementary Fig. 17). Consequently, the combination of PI and EGFP enabled two-color imaging, demonstrating the cost-effectiveness of a single-wavelength laser light source-based system with maintaining the coverage and the spatial resolution.

Taken together, even in simple microscopy configurations of descSPIM, where the detector size is smaller than the intended field of view, it is possible to achieve whole-brain 3D imaging by employing techniques such as tiling, registration, and stitching.

### Advanced applications in drug discovery and 3D pathology

We explored further the descSPIM's utility in wider biomedical research. We generated a 3D image of the entire tumor mass of a cancer cell line-derived xenograft (CDX) of BT-474 human breast cancer cell line, labeled with CD31-FITC (vessel marker) and PI, for application in drug discovery. Trastuzumab-DyLight™ 650 (an anti-cancer antibody drug targeting HER2) was pre-administered to the sample to visualize drug delivery inside the tumor mass in vivo. FA mode, TLS, and multi-directional imaging from 0° and 180° angles were utilized for the imaging.

The drug distribution associating the approximate regions from the vascular network was effectively visualized in the entire cancer tissue (Fig. 4a, Supplementary Fig. 18 and Supplementary Movie 4). The complete 3D data also depicted the non-homologous and biased drug delivery region by region inside the tumor mass (Fig. 4a, Supplementary Fig. 18 and Supplementary Movie 4). In this particular case, the peripheral regions of the tumor mass exhibited ineffective drug distribution, while the central regions received the drug more effectively. This observation was further supported by our quantitative analysis to figure out the distance of the drug-distributing area from the nearest vessels (Supplementary Fig. 19a −e and Supplementary Movie 5). The distances showed a long-tail type distribution, with the 95th percentile point having a distance of up to about 70 μm (Supplementary Fig. 19f, g). This result implies a diverse mechanism for transporting and distributing the administered anti-cancer drugs.

To additionally demonstrate the adoption of recent 3D pathology trends with clearing and light-sheet microscopy imaging[54], we also generated a 3D image of a 2 mm-thick CDX section with FA mode and TLS. The sample was stained with NHS-Alexa Fluor® 647 and PI in order to convert the acquired fluorescent images into a false-colored to mimic the pink and purple appearance of standard hematoxylin-eosin volume image (Fluo-HE[55,56]). The resultant data successfully reconstructed a 3D volume image of the entire Fluo-HE-colored tumor mass, albeit with a relatively low resolution for usual clinical pathology examination due to the low NA detection path (Fig. 4b and Supplementary Movie 6). Nevertheless, for future diagnostic objectives, descSPIM has the potential to reach not only the basic sciences but also the clinics.

These applications successfully demonstrated that descSPIM can function as a simple 3D imaging device in a wide variety of biomedical fields. The microscope's expanded potential, including a second illumination path, a wider FOV camera, and a higher magnification system, will make the advanced imaging and clinical applications easier and more straightforward.

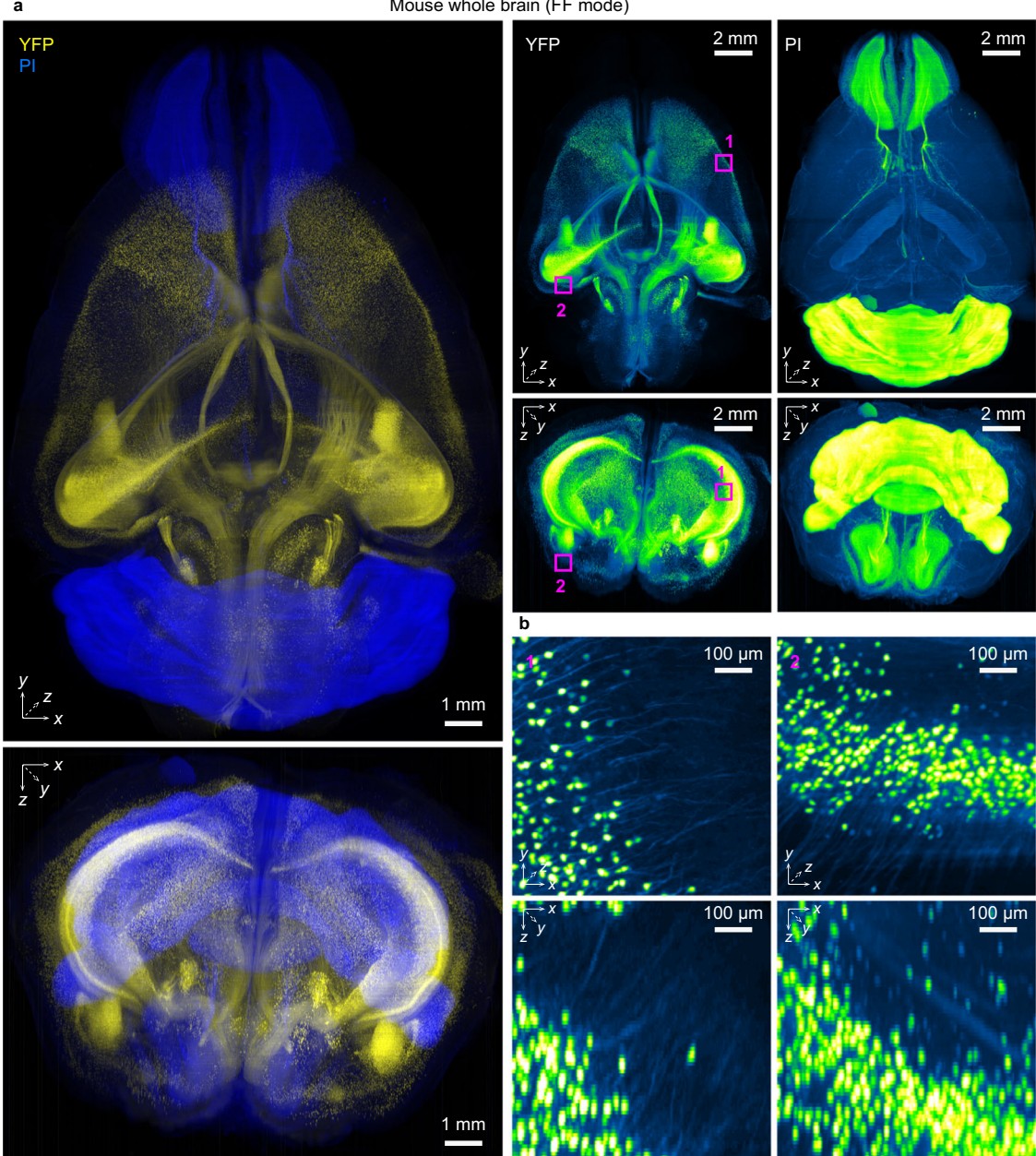

**Fig. 3 | Practical use of descSPIM in neuroscience. a** An example of propidium iodide (PI)-stained whole *Thy1*-YFP-H mouse brain imaging with full FOV (FF) mode and flat-field correction (FFC) adoption. PI and YFP were excited with 561 nm and 488 nm laser beams, respectively. Four image-stack tiles were collected from a single direction (angle 0°). Using the θ stage, a second set of four image tiles from the opposite direction (angle 180°) were then obtained. All the image tiles were stitched, aligned, and fused with BigStitcher and ANTs software. See Supplementary

Fig. 13 for the procedure. Voxel size: $3.45 \times 3.45 \times 10$ μm³. **b** Magnified images of YFP-expressing neurons within the dataset. Signal elongation along the axial direction was more modest in somas (10–20 μm in diameter) than in neurites (approximately 1 μm in diameter), consistent with the light sheet thickness (approximately 25 μm). Quantitative data is shown in Supplementary Fig. 14. The original grayscale 8-bit maps were pseudo-colored with a Blue, Yellow or Green Fire Blue look-up table. The imaging experiment was performed twice with nearly identical results.

## Dissemination of descSPIM in the research community

To facilitate the dissemination of descSPIM, we have conducted annual tutorial courses on its construction for end-users interested in tissue clearing techniques. Drawing inspiration from previous open-source initiatives such as openSPIM[45], mesoSPIM[38,40], diSPIM[29,30], and legolish/lemolish (https://www.microlist.org/listing/legolish/), as well as embracing the ethos of open microscopy[57], we have established a dedicated GitHub repository (https://github.com/dbsb-juntendo/descSPIM). This repository hosts comprehensive instructions for building descSPIM and fosters community engagement. As a result of these efforts, nearly 40 research groups worldwide have adopted

descSPIM or are considering to install, indicating its ability to address the growing demand for versatile tissue clearing applications (Fig. 5a). Indeed, descSPIM has been successfully applied in various biological contexts; imaging of a rat-brain coronal slice immunostained for tyrosine hydroxylase in the hypothalamus (Fig. 5b); visualization of a mouse brain hemisphere stained with SYTOX-Green (Fig. 5c); examination of ZsGreen-labeled tumor metastasis in mouse lung, stained with RedDot2 (Fig. 5d); observation of a stomach from Chat-Cre; R26-LSL-TdTomato; Dclk1-ZsGreen mouse strains (Fig. 5e); labeling of mouse intestine with PI (Fig. 5f); imaging of Tg(*fli1a:myr-EGFP*)$^{ncv2Tg}$ zebrafish expressing EGFP in endothelial cells (Fig. 5g); visualization of

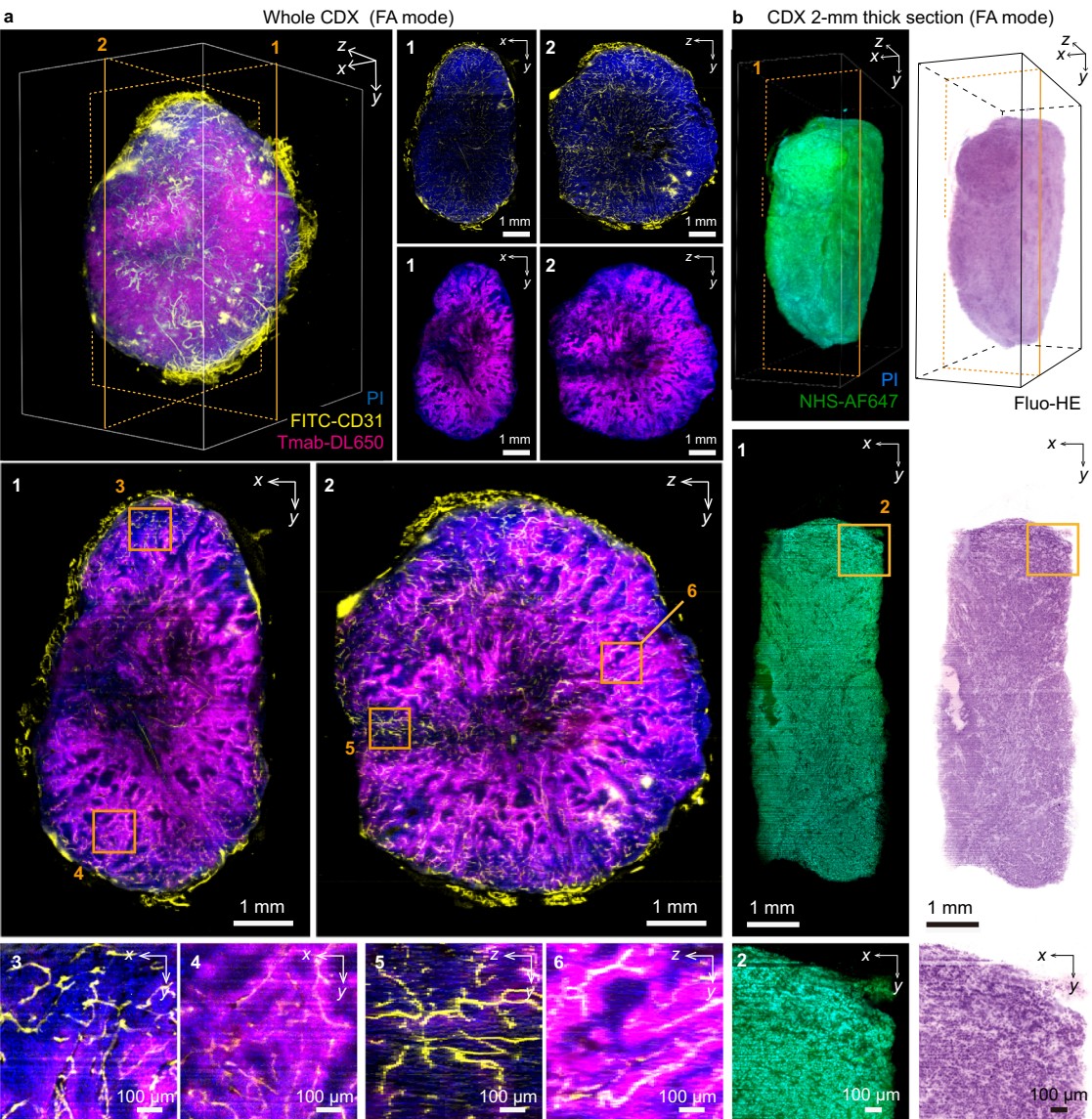

**Fig. 4 | Practical use of descSPIM in the drug discovery and 3D pathology. a** An example of Trastuzumab-administered whole cell-line derived xenograft (CDX; BT-474 human breast cancer cell line) imaging with fine axial (FA) mode, tilling light-sheet (TLS) method, flat-field correction (FFC) adoption, and multi-directional stack registration and fusing. Anti-CD31 antibody was also perfused before sampling. PI was stained during clearing. DL650 (Trastuzumab), PI, and FITC (CD31) were excited with 647 nm, 515 nm, and 488 nm laser beams, respectively. Using the θ stage, image tiles from the angles 0° and 180° were obtained. The two-sided stacks were then aligned and merged with ANTs and BigStitcher. Orthogonal cross-sections near the center of the specimen are shown as max intensity projection (MIP) images of a thickness of approximately 70 μm. Although validation of the imaging results by other methods is required, the descSPIM's potential to acquire apparent drug distributions is anticipated to be beneficial for drug discovery and development. Voxel size: 3.45 × 3.45 × 10 μm³. The original grayscale 8-bit maps were pseudo-colored with a Blue, Yellow or Magenta look-up table (LUT). The imaging experiment was performed at twice with nearly identical results. **b** An example of 2 mm-thick CDX imaging with FA mode, TLS method, and FFC adoption. The sample was stained with PI and NHS-Alexa Fluor® 647 before clearing, and false-colored hematoxylin and eosin colors (3D Fluo-HE) were applied. Alexa Fluor® 647 and PI were excited with 647 nm and 515 nm laser beams, respectively. The data demonstrates that descSPIM has the potential to be used routinely in 3D clinical pathology examinations. Voxel size: 3.45 × 3.45 × 10 μm³. The original grayscale 8-bit maps were pseudo-colored with a Blue, Green or Fluo-HE LUT. The imaging experiment was performed twice with nearly identical results.

z-stack images of E7.5 decidua labeled with SYTOX-Green (Fig. 5h); and 3D imaging of E12.0 RARE-lacZ whole mouse embryo (Fig. 5i). These data indicate that descSPIM is an affordable and easy-to-build microscope as well as a fully useful microscope.

## Discussion

Large-scale biological specimens now can be analyzed in three dimensions at the cellular levels or more using tissue clearing technologies. In this study, we proposed descSPIM as a low-cost and low-expertise LSFM system for tissue clearing end-users, by optimizing the fundamental concept of SPIM, which was initially proposed in 2004[16].

This device will facilitate the use of tissue clearing technologies in a broader range of biomedical fields more rapidly.

There were attempts at cleared sample imaging with the precedent open-sourced LSFM system (Supplementary Fig. 1). For example, openSPIM[45], a pioneer in open-source light-sheet microscopy, defaulted to a 10× high-magnification objective lens, which provides excellent performance in 3D live imaging of small animal embryos. A derived project then proposed a 4× objective design applicable to cleared tissue specimens[37]. Whole cleared mouse brain imaging has been demonstrated using this device, although apparently the resource does not assume to be open-sourced[58]. mesoSPIM, a recent

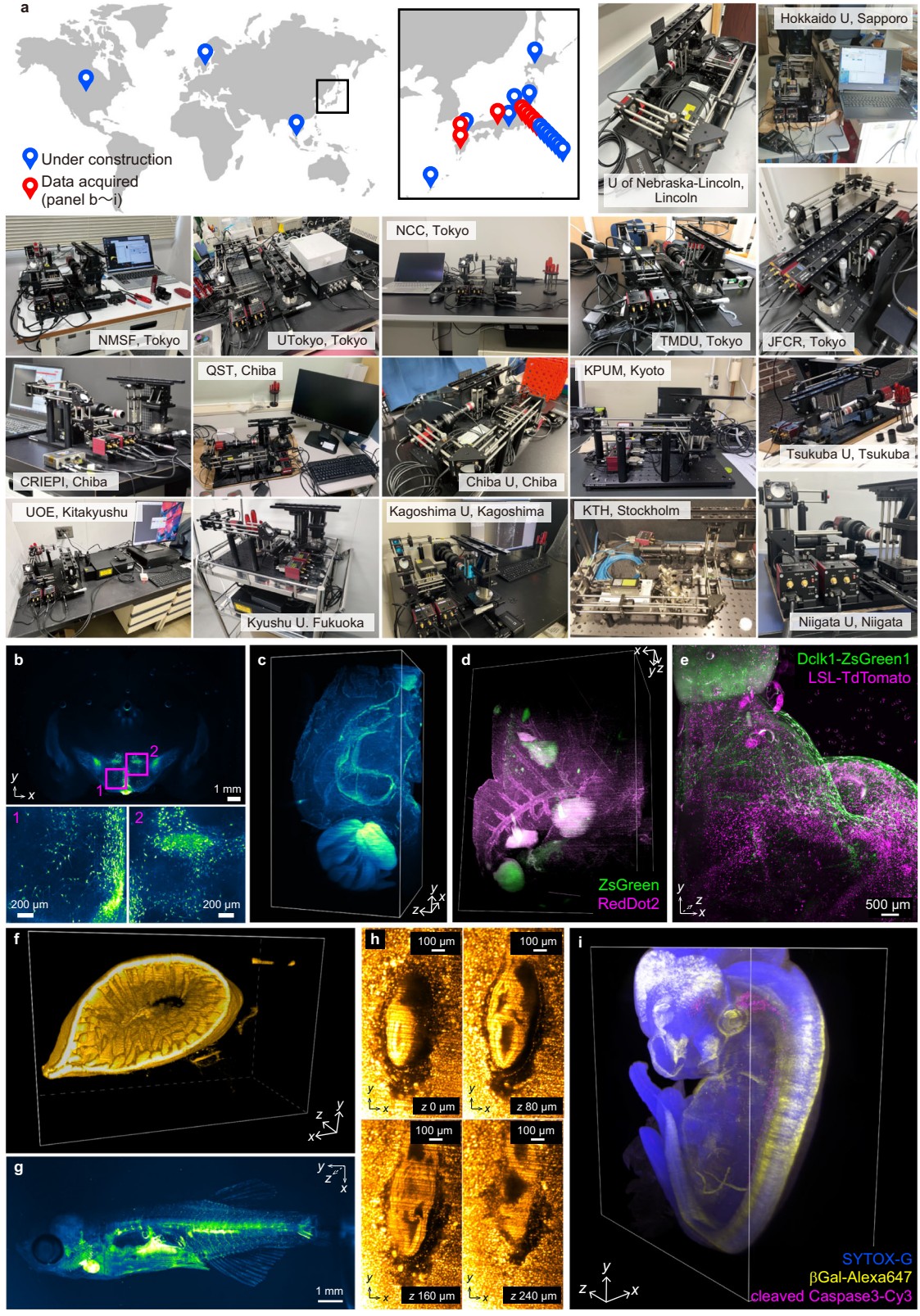

open-sourced LSFM system, further provided a specialized design for exhaustive 3D imaging of large-scale cleared specimens[38,40]. Although the system has sophisticated configurations, mesoSPIM was primarily designed with assuming operations at the microscopy facilities, resulting in thresholds for tissue clearing end-users who are unfamiliar with optics devices. While a recent Benchtop mesoSPIM was proposed to lower the cost of implementation, its construction is still difficult for biologists. In contrast to mesoSPIM/Benchtop mesoSPIM, the positioning of descSPIM is aimed to allow the end-users to utilize a personalized LSFM, next to their bench used for tissue clearing.

The descSPIM, despite its simplistic design and configuration, boasts remarkable versatility suitable for a broad spectrum of imaging applications essential for studying cleared tissues. To achieve this, descSPIM incorporates improvements in both new technical aspects

**Fig. 5 | Dissemination of descSPIM in the research community. a** Global distribution of descSPIM users and partial installation sites of the devices. CRIEPI: Central Research Institute of Electric Power Industry; JFCR: Japanese Foundation for Cancer Research; KPUM: Kyoto Prefectural University of Medicine; KTH: Royal Institute of Technology, Sweden; NCC: National Cancer Center Japan, Japan; NMSF: Nippon Medical School Foundation, Japan; QST: National Institute for Quantum Science and Technology, Japan; TMDU: Tokyo Medical and Dental University, Japan; UOE: University of Occupational and Environmental Health, Japan. **b** 3D image of a rat-brain coronal slice immunostained for tyrosine hydroxylase in the hypothalamus, acquired at Kagoshima University. *Xyz*: 6000 pix. × 3984 pix. × 189 slices. Voxel size: $3.0 \times 3.0 \times 10\ \mu m^3$. **c** 3D image of a mouse brain hemisphere stained with SYTOX-Green, acquired at QST. *Xyz*: 2160 pix. × 4096 pix. × 390 slices. Voxel size: $3.45 \times 3.45 \times 10\ \mu m^3$. **d** 3D image of ZsGreen-labeled tumor metastasis in mouse lung, stained with RedDot2, acquired at JFCR. *Xyz*: 2148 pix. × 3120 pix. ×

501 slices. Voxel size: $3.45 \times 3.45 \times 10\ \mu m^3$. **e** 3D image of a stomach from Chat-Cre; R26-LSL-TdTomato; Dclk1-ZsGreen mouse strains, acquired at Cancer Inst. *Xyz*: 2031 pix. × 3295 pix. × 521 slices. Voxel size: $3.45 \times 3.45 \times 10\ \mu m^3$. **f** 3D image of a mouse intestine with PI, acquired at CRIEPI. *Xyz*: 1512 pix. × 1011 pix. × 484 slices. Voxel size: $3.45 \times 3.45 \times 5\ \mu m^3$. **g** 3D image of Tg(*fli1a:myr-EGFP*)$^{ncv2Tg}$ zebrafish expressing EGFP in endothelial cells, acquired at NMSF. *Xyz*: 1795 pix. × 4096 pix. × 882 slices. Voxel size: $3.45 \times 3.45 \times 3.3\ \mu m^3$. **h** 3D image of E7.5 decidua labeled with SYTOX-Green, acquired at Kyushu U. *Xyz*: 856 pix. × 1468 pix. × 100 slices. Voxel size: $0.345 \times 0.345 \times 4\ \mu m^3$. **i** 3D image of E12.0 RARE-lacZ whole mouse embryo, acquired at KPUM. *Xyz*: 2048 pix. × 1080 pix. × 671 slices. Voxel size: $3.1 \times 3.1 \times 8.6\ \mu m^3$. The original grayscale 8-bit maps were pseudo-colored with a Blue, Green, Yellow, Magenta, Green Fire Blue, or Orange Hot look-up table. Each imaging experiments for cleared biological samples were performed at least once.

and existing systems, incorporating unique design features. In comparison to existing microscopes, descSPIM showcases pioneering technical solutions addressing specific challenges encountered in LSFM. Firstly, by eliminating the medium chamber and employing a two-stage synchronization approach, descSPIM enables 3D imaging, enhancing handling ease and accommodating various sample sizes and clearing reagents. Additionally, the streamlined process of capturing *xyt* images and reconstructing them into *xyz* images[44] significantly reduces imaging time, allowing for *z*-stack image acquisition within a few minutes, a remarkable improvement over traditional methods such as our advanced LSFM system[39], which typically took 15 minutes for a single *z*-stack throughout a mouse brain. Notably, the FF mode allowed for the efficient imaging of a whole mouse brain hemisphere in just minutes, ensuring a one-step completion of the process (Fig. 2a and Supplementary Fig. 12b). Switching to FA mode and tiling LSFM method achieved a spatial resolution of 4 µm in the lateral direction and 7 µm in the axial direction, allowing for nearly isotropic imaging of labeled nuclei (Fig. 2c). Moreover, the advanced implementations of the device for biomedical purposes, in conjunction with the full utilization of image processing techniques, demonstrated the successful data acquisition, visualization, and quantitative analysis of whole volumetric samples of the order of cm³ (Figs. 3, 4 and Supplementary Fig. 17). Remarkably, descSPIM's compact design, utilizing a minimal number of commercially available optical elements on a modest-sized breadboard, renders it suitable for personal use and particularly valuable in regions where access to high-end systems is limited.

For controlling this minimal LSFM, we initially proposed a method of simultaneously driving two different device-associated software. While this procedure enabled reproducible image acquisition, axial displacement for a few slides might occur. The required manual synchronization made it difficult to achieve complete elimination of human error, and in this case, post-correction of displacement of a few slides is easy. To automate the *z*-stack acquisition, a low-cost USB to IO expansion board was employed to synchronize camera initiation with stage movement. The latter, the custom software-based approach excels in reproducibility, but due to the setup complexity, we recommend prioritizing the utilization of the former, device-associated software for its simplicity.

However, the minimal layout also imposed inherent limitations on this initial basic system. For example, no optical devices to eliminate stripe shadow artifacts, which can occasionally cause issues in LSFM imaging, were equipped. When FA mode was employed, the restricted FOV along the *x*-axis led to operational inefficiency. We also utilized a manual $\theta$ stage along with single-side illumination and detection paths to enable imaging from various angles. This necessitated registration procedures to merge the acquired stacks. The ANTS software, which we employed for this procedure, has demonstrated reliability in several studies[59–61]. However, it is worth noting that the manual rotation implemented in the current descSPIM setup may introduce some

limitations in registration accuracy. In addition, an approach for imaging with a higher spatial resolution, particularly less than 1 µm resolution attained by other commercially available systems and advanced LSFM[33,36,62] has not yet been developed for this system. These shortcomings may be overcome by its expanded capability by using add-ons[18,50,63], while balancing the cost and complexity should also be considered.

We also highlight the educational value of descSPIM: the users' experience in constructing the basic equipment can assist in the acquisition of knowledge and alleviate the psychological burdens associated with advanced optics. Advanced microscopy systems are often treated as black boxes, and biological and medical researchers use them without knowing their visualization principles[18]. It is evident that an open-source project that generates sharing communities will significantly advance fluorescence bioimaging technology. The widespread adoption of descSPIM is thus anticipated to result in additional advanced biomedical discoveries spurred by tissue clearing technologies.

## Methods

All experimental procedures and housing conditions of the animals were approved by the Animal Care and Use Committees of Juntendo University (1569-2022279 and 1372-2022211), National Cancer Center Research Institute (T21-012), Brain Research Institute, Niigata University (SA01266), National Institute for Physiological Sciences (22A044), Kagoshima University (VM21058), the National Institutes for Quantum Science and Technology (23-1032; R5-10-1), the Japanese Foundation for Cancer Research ((10−01−22)), the University of Tokyo (A2023M095), Central Research Institute of Electric Power Industry (2305), the Nippon Medical School (2022-020), Kyushu University (A22-029-4), and Kyoto Prefectural University of Medicine (M2023-178). All of the animals were cared for and treated humanely in accordance with the Institutional Guidelines and with the recommendations of the United States National Institutes of Health for experiments using animals.

### Tissue samples for specification evaluations and initial application
We used 8-week-old male C57BL/6N mice (Japan SLC, Inc., Shizuoka, Japan) (for Fig. 2a, c), 8- and 14-week-old male C57BL/6N mice (Japan SLC, Inc.) (for Supplementary Fig. 12b), 9-week-old female *Thy1*-YFP-H Tg mice (B6.Cg-Tg(*Thy1*-YFP)HJrs/J, The Jackson Laboratory, Identifier: 003782) and 35-week-old female *Thy1*-GFP-M Tg mice (STOCK Tg(*Thy1*-EGFP)MJrs/J, The Jackson Laboratory, Identifier: 007788)[51] (for Fig. 3a, Supplementary Fig. 16 and 17). C57BL/6N mice were sacrificed immediately after purchase without prior housing in the local animal facility. *Thy1*-mice were housed in cages at 20-24 °C, 30−65% humidity in a 12-hr light/dark cycle, with ad libitum food and water.

To establish cancer cell line–derived xenograft (CDX) models (Fig. 4a, b), the mixture of BT-474 human breast cancer cells (American

Type Culture Collection, Manassas, USA, HTB-20; $1.0 \times 10^7$ cells) suspended in PBS and equal volume of Matrigel® (growth factor reduced and phenol-red free type, Corning, NY, USA) was subcutaneously injected into the flank of the 5 to 6-week-old female SCID-beige mice (CB17.Cg-Prkdc[scid]Lyst[bg-J]/ CrlCrlj, Charles River Laboratories, Kanazawa, Japan). DyLight[TM] 650-labeled Trastuzumab was prepared as follows: Dialyzed Trastuzumab (Herceptin®, Chugai Pharmaceutical, Tokyo, Japan) using 10 K MWCO Slide-A-Lyzer dialysis cassette (Thermo Fisher Scientific, OR, USA, #66810) and DyLight[TM] 650 (Thermo Fisher Scientific, #62265) were incubated at a ratio of 1:10 for 1 h with light protection. Sodium bicarbonate (Thermo Fisher Scientific, #J60408-AK) was added to keep the pH at an alkaline range to accelerate the reaction. After the reaction, unreacted dye was removed using PD-10 desalting column (Cytiva, MA, USA, #17085101). Concentrations of DyLight[TM] 650-labeled Trastuzumab were measured by Nanodrop 2000 (Thermo Fisher Scientific) and the molar fluorescent per protein ratio was calculated according to supplier's instruction. The fluorescent-labeled Trastuzumab was stored at 4 °C until use. For in vivo study, FITC-labeled anti-murine CD31 antibody (BioLegend, CA, USA, #102506) and the DyLight[TM] 650-labeled Trastuzumab were intravenously administered from the tail vein 24 h before sacrifice at a dose of 2.5 mg/kg and 4 mg/kg, respectively.

Mice were housed in a specific pathogen-free room at $22 \pm 0.5$ °C and $55 \pm 10\%$ humidity, with a 12-h light/dark cycle. In the protocol approved by the Ethics Committee for Animal Experiments at National Cancer Center Research Institute, the humane endpoint for tumor volume was defined as a volume of 2000 mm³ or a tumor volume not exceeding 10% of body weight. Tumor volume and body weight were measured at least twice a week to ensure that the humane endpoint was not reached during animal experiments were conducted.

For sampling the tissues, the mice were sacrificed under the deep anesthetization with intraperitoneal administration of an overdose (>100 mg/kg) of pentobarbital (pentobarbital sodium salt, nacalai tesque, Kyoto, Japan, #02095-04) or inhalation of isoflurane (Viatris Inc., PA, USA, #901-036504) and then transcardially perfused with PBS (occasionally supplied with -10 U/ml of heparin) and 4% paraformaldehyde in PBS. Then, the tissue samples were excised and postfixed with the same fixative for 8–24 h at 4 °C. The specimens were used for the subsequent clearing procedure or stocked at 4 °C in PBS with NaN₃ until required for use. For the vascular labeling with Evans blue, a 14-week-old male C57BL/6N mouse was anesthetized and injected intravenously with 100 μL of Evans blue (20 mg/mL dissolved in physiological saline and filtered with a 0.22 μm filter) (Tokyo Chemical Industry (TCI), Tokyo, Japan, #E0197). 10 minutes after injection, an additional 10 mL of Evans blue solution (0.2 mg/mL) was administered, followed by heparinized PBS and 4% PFA (10 mL each) transcardially perfused as above. Post-fixed hemisphere samples were processed using 60–100% methanol and BABB, following the same clearing protocol used for TO-PRO®-3-stained brain tissue as described in the following section.

## Tissue clearing and nuclear staining

CUBIC tissue clearing was performed according to our previous protocols[39,44] with some optimizations for each sample. We used commercialized CUBIC-L and CUBIC-R + (N) reagents (TCI, #T3740 and #T3983). For PI staining, CUBIC-L-treated specimens were immersed in HEPES-NaCl buffer (10 mM HEPES: TCI, #H0396; 500 mM NaCl: TCI, #S0572; 0.05% NaN₃: nacalai tesque, #31208-82) with 2–3 μg/mL Propidium iodide (PI) (DOJINDO, Kumamoto, Japan, #343-07461) at 37 °C for 3–7 days according to the sample size. For Fluo-HE staining, the published protocol[55] was used for some modifications. The CUBIC-L-treated specimen was immersed in ATS buffer (50 mM sodium acetate/acetic acid buffer, pH 5: nacalai tesque, #311-18 and TCI, #A3377; 30% tetrahydrofuran: TCI, #T0104; 500 mM NaCl) containing 0.5 or 0.25 μg/mL NHS-Alexa Fluor® 647 (Thermo Fisher Scientific, #A37573)

and 2 μg/mL PI at 4 °C for one week, and then cleared with CUBIC-R + (N). The sample was embedded in CUBIC-R-agarose[44] (http://cubic.riken.jp/) using glass cuvettes (GL Science, #F10-G-10 or #F10-G-20) or plastic cuvettes (Bio-Rad, #1702415) for imaging.

BABB clearing and TO-PRO®-3 staining were performed as follows. Post-fixed hemisphere samples were incubated with 10% (vol/vol) 1,2-hexanediol (TCI, #H0688) in Milli-Q water at room temperature for 1 day. The delipidated samples were then immersed in PBS containing 500 mM NaCl, 10% (vol/vol) ethanol, and 1 μM TO-PRO®-3 iodide (Thermo Fisher Scientific, #T3605) at room temperature for 4 days. After washing with PBS, the samples were immersed in 60% and 80% methanol (FUJIFILM Wako Pure Chemical Corporation, Osaka, Japan, #137-01823,) for 4 h at each step, twice in 100% methanol overnight and then for 8 h, and finally in BABB (a 1:2 mixture of benzyl alcohol and benzyl benzoate; TCI, #B2378 and #B0064) overnight at room temperature. The sample was imaged using glass cuvettes (GL Science, #F10-G-10) while submerged in BABB.

## Microscopy construction

All parts list and the detailed information on the construction and usage of descSPIM are opened in our GitHub website (https://github.com/dbsb-juntendo/descSPIM). In brief, four-color laser lights (Cobolt, Skyra, 488 nm/50 mW, 515 nm/50 mW, 561 nm/50 mW, and 647 nm/50 mW) were coupled into a single-mode fiber (Schäfter +Kirchhoff, PM series, effective NA = 0.072) and introduced to the excitation optical path. An achromatic convex lens (Thorlabs, C254-100-A, $f = 100$ mm) and two selectable cylindrical lenses (Thorlabs, LJ1144RM-A, $f = 500$ mm plano convex for the full-FOV mode; Thorlabs, ACY254-150-A, $f = 150$ mm achromatic for the fine-axial mode) were used for collimation and light-sheet formation, respectively. A CMOS camera (Thorlabs, S895MU, $7.45 \times 14.1$ mm² imaging area, $3.45 \times 3.45$ μm² pixel size, and 8.9 Megapixels) with a tube lens (Thorlabs, TTL100-A, $f = 100$ mm) and a 2× objective lens (Thorlabs, TL2X-SAP, NA = 0.1, WD = 56.3 mm) were used for the detection optical path. One of Φ25 mm bandpass filters (Thorlabs, FBH520-40, 500–540 nm, FBH550-40, 530–570 nm, FBH600-40, 580–620 nm, or FBH700-40, 680–720 nm) or a longpass filter (Thorlabs, FBLH0550, 550 nm) was placed in the detection path between the objective and the tube lenses. Motorized translation stages (Thorlabs, XR25P/M and Z825B) were adopted under the detection path and the sample stage. The sample in an appropriate cuvette was placed on a suspended sample stage combined with a $\theta$ stage (Thorlabs, PR01/M). No oil chamber was used. $X$ and $y$ positions were manually adjusted by manual translation stages (Thorlabs, XR25P/M for fine $x$ positioning, 1519/M and MVS05/M for rough and fine $y$ positioning, respectively) before imaging.

The CAD design of mechanical components was performed in Autodesk® Fusion 360 software.

## General imaging procedure

descSPIM imaging was performed with one-sided illumination. 8-bit images were collected by scanning the sample in the $z$-direction as a movie file, the method inspired by MOVIE scan[44]. Even though the camera's maximal bit depth is 12-bits, we recommend 8-bit data collection to prevent I/O delays during movie imaging on a PC with midrange specifications. During the imaging procedure, the focus was maintained by two motorized stages that traced synchronously (Supplementary Fig. 3). The synchronous speed correction value (the relative velocity of actuators) was theoretically calculated as follows: The focal length of the objective lens ($f$) can be expressed as a linear combination of the ratio of physical distance ($D$) to the refractive index of the optical path ($n$).

$$f = \frac{D_1}{n_1} + \frac{D_2}{n_2} + \cdots \tag{1}$$

In the initial state of descSPIM, $f$ can be described as;

$$f = \frac{A}{n} + \frac{B}{1.00} \qquad (2)$$

where $A$ and $B$ are the distances described in Supplementary Fig. 3. When the sample position and the objective lens ($z_{sample}$ and $z_{detect}$) are moved in opposite directions ($dz_{sample}$ and $dz_{detect}$), $f$ can be described as:

$$f = \frac{A - dz_{sample}}{n} + \frac{B + dz_{sample} - dz_{detect}}{1.00} \qquad (3)$$

Therefore, $dz_{detect}$ can be expressed as a function of $dz_{sample}$, where

$$dz_{detect} = \frac{n - 1.00}{n} dz_{sample} \qquad (4)$$

If CUBIC-R ($n = 1.52$) is utilized for clearing;

$$dz_{detect} = 0.342 \times dz_{sample} \qquad (5)$$

Therefore, the synchronous speed correction value is 0.342 in this case. This means that if the sample stage is moving at 100 µm/sec, the detection optics should be moving at 34.2 µm/sec. In the practical situation, however, the synchronous speed correction value can be influenced by the actuators' capacity for weight and the PC's specifications, resulting in a deviation from the theoretical estimate. We determined the practical correction value using the method illustrated in Supplementary Fig. 6.

When obtaining a $z$-stack image of a mouse hemisphere with FF mode (Fig. 2a), the velocity of the sample stage ($v_{stage}$) was set to 50 µm/sec and the velocity of the detection optics ($v_{detect}$) to 17.3 µm/s on the actuator-associated software Kinesis® (Thorlabs). The traveling distance of the sample stage was 10 mm ($dz_{stage} = 10$ mm, $dz_{sample} = 3.46$ mm). The time-lapse image was obtained with the exposure time of 200 ms. The resulting $xyt$ image was converted to $xyz$ format with a $z$-range of 10 mm and a $z$-interval ( $= v_{stage} \times$ exposure time) of 10 µm using ImageJ/Fiji[64]. Due to the specification of the camera-associated software ThorCam™ (Thorlabs), files were split when they exceed 1 GB. Therefore, a final stack was generated using ImageJ/Fiji's concatenate function. As for a mouse coronal section in FA mode (Fig. 2b), $v_{stage}$ was set to 25 µm/sec and $v_{detect}$ to 8.5 µm/sec. The traveling distance of the sample stage was 10 mm ($dz_{stage} = 10$ mm, $dz_{sample} = 3.40$ mm). The time-lapse images of multiple stacks for TLS were obtained with the exposure time of 200 ms. The resulting $xyz$ images had a z-range of 10 mm and a z-interval of 5 µm. To visualize the fluorescent signals from PI, the 515 nm excitation laser light (3 mW output) and the 550 nm longpass filter were selected for measurements. We used a notebook PC (Intel® Core™ i7-8750H, 16GB RAM, SSD 256 GB M.2 2280 S3-M SDAPNUW-256G) for the operation and the initial data processing.

In addition, a low-cost USB to IO expansion board (Arduino Uno Rev3) was employed to synchronize camera initiation with stage movement to automate $z$-stack acquisition (Supplementary Fig. 10). A simple Jupyter Notebook was developed to guide users through a $z$-stack acquisition (available on GitHub). The camera was controlled in µManager, and the stages were controlled with a napari-micromanager widget.

To collect comparative data with an advanced system (Supplementary Fig. 16), the PI-stained whole *Thy1*-YFP-H mouse brain was imaged with our advanced LSFM as previously reported[39]. Images were acquired as 16-bit tiff files. Total six stacks per channel were collected

with the system-equipped TLS procedure. After tiling, the resulting stacks were finally converted to 8-bit data for visualization.

## PSF evaluation
Fluorescently-labeled beads (Nile Red, 1 µm diameter; Thermo Fisher Scientific) were diluted in the CUBIC-R + (N) agarose gel (1:1000, v/v) and filled into a cuvette for measurements. The fluorescent images of the Φ1 µm beads had a pixel size of 3.45 µm in both FF and FA mode. We acquired 300-µm-thick $z$-stacks at 10 µm intervals in FF mode and 120-µm-thick $z$-stacks at 3 µm intervals in FA mode, and used them to reconstruct $xyz$ images of the beads. To visualize the fluorescent signals from the beads, the 515 nm excitation laser light (5 mW output) and the 550 nm longpass filter were selected for measurements. The FWHM values were evaluated by fitting their fluorescence intensity profiles around the central peak intensity using Gaussian functions. To avoid evaluation of intensity profiles derived from multiple fluorescent beads or position-dependent PSF elongations, we selected 10 of the dozens of FWHM values measured in a certain region of interest from the smallest values and evaluated their means and standard deviations.

Furthermore, we assessed the point spread functions (PSFs) in FA mode using organic solvent-based clearing reagents, specifically BABB and ECi. Quantum dots (Qdot™ 605 ITK™ Amino (PEG) Quantum Dots; Thermo Fisher Scientific) were diluted (1:100, v/v) in 2% agarose (w/v) + 2% gelatin (w/v, #16631-05, Nacalai) gel. The gel underwent dehydration with ethanol (50% for 2 h, 70% for 2 h, 100% for 2 h, 100% overnight), followed by refractive index adjustment using either BABB or ECi as the organic solvent. Subsequently, the gel containing the quantum dots was placed into a cuvette for measurement. The fluorescent images of the quantum dots had a pixel size of 3.45 µm. We acquired 150-µm-thick $z$-stacks at 3 µm intervals and used them to reconstruct $xyz$ images of the beads. To visualize the fluorescent signals from the quantum dots, the 561 nm excitation laser light (7 mW output) and the 580–620 nm bandpass filter were selected for measurements.

## Light-sheet shape evaluations
Excitation light sheets propagating before and after the focus of the cylindrical lenses were measured by a beam profiling CCD camera (Ophir, SP920s). The 515 nm laser light was used for evaluations. The beam profiler was placed at the sample position, and the intensity profiles were measured before and after the focal point, where the FWHM value of the intensity profile in the $z$-direction was the smallest. The beam profiles were measured at equal intervals (1 mm for FF mode and 0.1 mm for FA mode) until the $x$-position, where the widths were enlarged by √2. Since the FWHM values plotted against $x$-position gave an asymmetric shape due to spherical aberration, the Rayleigh lengths were calculated by fitting with the fifth-order polynomials.

## Adoption of the flat-field correction
The flat-field correction was incorporated from the literature[50] and implemented as an ImageJ/FIJI macro. Briefly, 1 µM fluorescein was added to the CUBIC-R reagent in a cuvette, and an image (reference image) was obtained using the same illumination mode (FF or FA) as the sample imaging except for exposure time of 2 s and bit-depth of 12-bit. Then, the median values in the $y$ direction of the reference image were obtained and utilized to calculate the coefficient values for normalizing each median value to the peak median value in the $y$ direction. The coefficient values were then applied to the sample stack in order to equalize the intensities along the $y$-axis.

## Adoption of the tiling light-sheet method
Tiling light-sheet imaging methods (TLS) with FA mode (Figs. 2c and 4a, b) was adopted from literature[19] so that the eFOV (2× Rayleigh length) of the sheet illumination can cover the wider area along the $x$-axis after tiling. With a manual linear translation stage (Thorlabs, CTA1/

M) affixed to the FA mode cylindrical lens (f = 150 mm), the light-sheet focus positions were moved along the $x$-axis by 500 μm in physical length (approximately 760 μm in CUBIC-R, RI = 1.52). In order to acquire multiple stacks at various light-sheet focus positions, the initial focal position was aligned at the center (or edge) of the sample. After collecting the first $z$-stack, the stage was moved 500 μm away and a second $z$-stack was collected, yielding $z$-stacks with focal points displaced by distance × refractive index value (less than twice the Rayleigh length in FA mode). These measurements were repeated to derive $z$-stacks for all of the sample's regions of interest.

Tiling position in the $x$-axis was identified using a custom ImageJ macro executing the process as follows: (1) A part of resliced $x$-$z$ images from acquired $z$-stacks that had different sheet focus positions were prepared. (2) The "Bandpass Filter" function of ImageJ was applied to a 50 × 100-pixel kernel. (3) A standard deviation (SD) was calculated for the resultant kernel area image. If the area contains axially focused high-contrast signals, the SD increases. (4) The SDs of the same kernel position from the two resliced $x$–$z$ images were compared. (5) The point in $x$ where the two SDs were inverted was identified as the tiling $x$-coordinate. Finally, the tiling was completed using the "combine" function of ImageJ at the calculated $x$-coordinate or using the fusion function of BigStitcher[53] with a 100-pixel overlap.

## Multi-positional and multi-directional imaging

Whole *Thy1*-YFP-H mouse brain imaging with FF mode (Fig. 3) was performed as follows: Each $z$-stack covering about two-thirds of the hemisphere area was acquired from both the dorsal (angle 0°) and ventral (angle 180°) sides, respectively. Since the whole mouse brain area could be covered with four stacks per direction, we collected a total of eight stacks from the two opposite directions. Image intensity was corrected by FFC. Then, the four image stacks per direction were stitched with BigStitcher[53] to reconstitute a single stack covering the whole brain. Then, the stacks from the dorsal and ventral sides were registered by ANTs software[52] (details are described in the following section) and fused with BigStitcher in order to complement the degraded parts of the images. The procedures were overviewed in Supplementary Fig. 13. As for whole *Thy1*-GFP-M mouse brain imaging with FF mode (Supplementary Fig. 17), four $z$-stacks covering about two-thirds of the hemisphere area were acquired from the dorsal side. Image intensity was corrected by FFC. Then, the four image stacks were stitched with BigStitcher to reconstitute a single stack covering the whole brain.

The CDX imaging data in Fig. 4a was also collected from the two opposite directions (angles 0° and 180°) with TLS. They were registered by ANTs and fused with BigStitcher after the TLS tiling, similar to the whole brain case.

## General image processing for visualization

Acquired images were processed, reconstituted, and visualized as a false-colored image by using Fiji and NIS Elements (Nikon, version 5.02). The intensity was adjusted with the brightness and contrast functions. Occasionally, some images were cropped, and their orientations were adjusted to prepare figure panels. The false-colored to mimic the pink and purple appearance of standard HE image (Fig. 4b) was performed according to the published literature[56] and by using an open-source Python code (https://github.com/dbsb-juntendo/descSPIM).

## Registration, fusion and stitching of multiple stacks

We used ANTs[52], custom python code, and BigStitcher on ImageJ/FIJI (53) for the image registration, fusing, and stitching taken from bidirectional angles (0° and 180°) and different field of views, respectively. The overview of this procedure is explained in Supplementary Fig. 13. The related code has been shared on our GitHub repository (https://github.com/dbsb-juntendo/descSPIM).

The registration and fusing of two corresponding stacks obtained from bidirectional angles (0° and 180°) were conducted as follows (Supplementary Fig. 13d). The flat-field correction was applied to all the stacks before processing. The orientation of angle 180° stacks was matched to that of angle 0° stacks using the "Flip Z" and "Flip Horizontally" commands in ImageJ/FIJI. Then, all the stacks were compiled into full-sized and downsized NIfTI format stacks. In the downsized stacks, the voxel spacing was reduced by 50% in the $xy$-axis direction and retained one in every four slices in the $z$-axis direction. This downsizing was crucial for successful ANTs registration calculations to avoid memory handling errors, especially when handling large data (about 10 GB).

The registrations were conducted separately for nuclear channel stacks and target signal channel stacks. For the nuclear staining channel, the downsized NIfTI volumes were used to generate a transformation matrix through ANTs deformable registration (rigid-affine-SyN). The NIfTI volume files from angle 180° were registered with their corresponding NIfTI volume files from angle 0°. The transformation matrix was then applied to the original, full-size 180° angle NIfTI file. The full-size angle 0° stack and the registered angle 180° stack were then averaged to produce a complemented stack.

For the target channel stacks (the YFP channel in our case), global and local transformation matrices were sequentially generated. The global transformation matrix was computed by applying an affine registration of the downsized angle 180° NIfTI stack of the nuclear channel with its angle 0° counterpart. The obtained affine transformation matrix was then applied to the downsized angle 180° NIfTI stack of the target channel. Further, using ANTs deformable registration (rigid-affine-SyN), the local transformation matrix for the affine-transformed angle 180° stack to be registered to the downsized angle 0° stack of the target channel was yielded. These two matrices were successively applied to the original, full-size angle 180° stack of the target channel, and the full-size angle 0° stack and the registered angle 180° stack were finally averaged to produce a complemented stack.

We compared registration accuracy by using different levels of downsized stack to obtain transformation matrices: 25%-downsize (assuming a lab-equipped laptop PC) and 50%-downsize (assuming a high-spec workstation or a supercomputer) (Supplementary Fig. 14a). At lower magnification, the registered images using both 25% and 50% downsizing appeared to be well-registered (Supplementary Fig. 14b). At higher magnification, however, there are some nuclei or neurons that were duplicated or overlapped in the registered image using 25%-downsize (Supplementary Fig. 14c). In the registered image using 50%-downsize, these duplications and overlappings were improved (Supplementary Fig. 14c). The registration process with 50%-downsized stacks was successfully executed at our lab-equipped high-end workstation PC (CPU: Intel® Dual Xeon Gold 6348 Ice Lake-SP 28Cores/56Theads 2.6 GHz/3.5 GHz L3 = 42MB TDP235W DDR4-3200, GPU: NVIDIA® A100 80GB PCLe 4.0, RAM: 1024GB (64GB x 16) ECC Registered DDR4-3200, the calculation times are summarized as Table 1). The process stopped raising memory allocation errors with a decent laptop PC (64 GB RAM, 8 cores, 20 threads) when applying the transformation matrix to the original full-size images (~10 GB).

Finally, the registered and fused stacks encompassing the entire brain were stitched together using BigStitcher (Supplementary Fig. 13b). To conserve computational resources, we cropped each stack by 180 pixels in the $x$-axis and 1150 pixels in the $y$-axis (from the original shape of 2160 × 4096 pixels) without affecting the precision of the manual stitching procedure in BigStitcher. The raw images of each tile were composed of TIFF stacks with a lateral resolution of 3.45 μm, an axial resolution of 10 mm, and consisted of 997 slices. The data size of each tile was 8.8 GB. The whole mouse brain image was reconstructed from these eight tiles as described, resulting in a final data size of 22.4 GB.

**Table 1 | The calculation times for registration of propidium iodide (PI)-stained Thy1-YFP-H mouse brain images**

| | Nuclear staining channel | | Target channel | | | |
|---|---|---|---|---|---|---|
| ANTs | SyN registration | Apply transform | Affine registration | Apply affine transform | SyN registration | Apply transform |
| Channel | Nuclear staining | Nuclear staining | Nuclear staining | Target | Nuclear staining | Target |
| Resolution | Downsize (50%) | Full size | Downsize (50%) | Downsize (50%) | Downsize (50%) | Full size |
| FOV1 | 5 h 53 m 10 s | 11 m 39 s | 7 m 53 s | 56 s | 5 h 46 m 30 s | 10 m 19 s |
| FOV2 | 5 h 53 m 50 s | 12 m 43 s | 7 m 01 s | 1 m 02 s | 5 h 52 m 30 s | 11 m 22 s |
| FOV3 | 5 h 53 m 30 s | 14 m 05 s | 9 m 43 s | 1 m 12 s | 5 h 46 m 30 s | 12 m 26 s |
| FOV4 | 5 h 53 m 40 s | 12 m 11 s | 12 m 52 s | 1 m 02 s | 5 h 46 m 50 s | 11 m 10 s |

## Quantitative evaluation of registration accuracy

We calculated two types of evaluation indices, normalized Mutual Information (MI) and normalized Zero-Means Cross-Correlation (ZNCC), to assess the registration quantification between the two images (Supplementary Fig. 14d). These metrics were selected for their sensitivity to intensity variations and structural alignment, which is crucial for our analysis.

The formulas employed are as follows:

Mutual Information

$$MI(X,Y) = H(X) + H(Y) - H(X,Y) \tag{6}$$

$$normalized\ MI(X,Y) = \frac{MI(X,Y)}{\sqrt{H(X) \times H(Y)}} \tag{7}$$

$X$, $Y$ are the pixel intensity arrays of a corresponding slice in a pair of images. $MI(X, Y)$ is the normalized mutual information for $X$ and $Y$, $H(X)$ and $H(Y)$ are the entropies of $X$ and $Y$, and $H(X, Y)$ is the joint entropy for $X$ and $Y$.

Zero-Means Cross-Correlation:

$$ZNCC(X,Y) = \frac{\sum(X - \mu X) \times (Y - \mu Y)}{\sum(X - \mu X)^2 \times \sum(Y - \mu Y)^2} \tag{8}$$

$X$, $Y$ are the pixel intensity arrays of a corresponding slice in a pair of images, $\mu X$ and $\mu Y$ are the means of the images $X$ and $Y$.

The calculation and graph preparation were implemented with our custom Python code. Registration accuracy improvement was confirmed quantitatively with these metrics.

## Quantitative analysis of the drug distribution distance from blood vessels

For pre-processing, FITC-labeled CD31 images (CD31-FITC) and DyLight™ 650-labeled trastuzumab images (Tmab-DL650) of CDX samples were processed with ImageJ/FIJI. For CD31-FITC, a background subtraction of ImageJ with a radius of 4 was adopted. Then, max intensity projection (MIP) images from every 10 slices were created. For binarization, an auto-thresholding value was determined by applying the Otsu method to the central MIP slice (slice number = 34). The threshold value was applied to all slices. For Tmab-DL650, background subtraction of ImageJ was executed with a radius of 50, followed by binarization in a similar process. Drug-associating or non-associating vessels (Supplementary Fig. 19d) were selected based on colocalization of the binarized signals.

We determined the drug distribution distances by our custom code to search for the nearest vascular point from the drug distribution contour and then figuring out the mean drug distribution distance for each of the vascular points. Binarized images of CD31-FITC and Tmab-DL650 from the corresponding MIP slice were used for the process. The drug distribution contour was obtained using an OpenCV module in Python. The nearest point on the CD31-FITC-labeled vascular contour from each point on the belonging drug contour was determined with a kd-tree algorithm for rapid nearest-neighbor search (https://docs.scipy.org/doc/scipy/reference/generated/scipy.spatial.cKDTree.html). The distance line drawn between the designated points of the vessel and drug contours was considered valid in the case that the whole line was embraced inside the drug distribution area. On the contrary, if the line traversed an area outside the drug distribution area, the line was omitted (Supplementary Fig. 19a, left panel). Occasionally, multiple points on the drug contour that are nearest to a specific point on the vessel contour can be selected (i.e., drug distribution distance from a specific vessel is expressed in a range of values). In this case, their average was calculated as the final value of the drug distance (Supplementary Fig. 19a, right panel). Finally, the lines were overlaid on the binarized images for visualization (Supplementary Fig. 19b–d and Supplementary Movie 5). The slice-wise quantitative results were presented as boxplots of drug-to-vessel distances (Supplementary Fig. 19e, f), and a histogram (Supplementary Fig. 19f).

## descSPIM installed in KTH Royal Institute of Technology

For Fig. 5a, the system installed in KTH Royal Institute of Technology was built according to the guide provided on the descSPIM Github with some modifications (collimating lens: Thorlabs #AC254-050-A, detection objective: Zeiss EC Plan-NEOFLUAR 5×0.16 NA, tube lens: Thorlabs #AC254-100-A, camera: CM3-U3-31S4M-CS 1/1.8" Chameleon®3 Monochrome Camera).

## descSPIM installed in Kagoshima University

For Fig. 5a, the system installed in Kagoshima University was built according to the guide provided on the descSPIM Github with some modification (camera: Nikon DS10 controlled with NIS elements Ar, tube lens: Thorlabs TTL200-A), laser source: Cobolt Skyra 488 nm/561 nm/647 nm). For Fig. 5b, A 1-mm-thick slice of rat brain (Sprague-Dawley, male) was delipidated with CUBIC-L (TCI, #T3740) for 3 days at 37 °C, followed by immunostaining with normal donkey serum (1:50, Jackson Immuno Research lab., West Grove, PA, 017-000-121) and anti-tyrosine hydroxylase antibody (1:1000, Merck, Darmstadt, Germany, MAB318) in PBS containing 10% TritonX-100 for 3 days at 4 °C. After overnight wash, the sample was incubated with Alexa Fluor 488-conjugated goat Fab for anti-mouse IgG1 Fc-specific (1:1000, Jackson Immuno Research lab., 115-547-185) in PBS containing 10% TritonX-100 for 3 days at 4 °C. The stained sample was then RI-matched with SeeDB2[65]. The z-stack data (approximately $18 \times 12 \times 1.9$ mm³, voxel size: $2.98 \times 2.98 \times 10$ μm³) were collected using time-lapse acquisition mode. The panels were prepared with ImageJ.

Rats were housed in cages at 23 °C and 30–70% humidity in a 12-hr light/dark cycle, with ad libitum food and water.

## descSPIM installed in National Institutes for Quantum Science and Technology

For Fig. 5a, the system installed in National Institutes for Quantum Science and Technology (QST, Japan) was built according to the guide

provided on the descSPIM Github with some modifications (equipped laser light source: Cobolt, Skyra, 488 nm/50 mW and 647 nm/50 mW). For Fig. 5c, a tauopathy model mouse brain (female) was delipidated with CUBIC-L for 5 days at 37 °C, followed by the whole-mount staining with SYTOX-Green (ThermoFisher Scientific, #S7020)[39]. The sample was then RI-matched with CUBIC-R. The z-stack tiff data (approximately 7.0 × 14 × 6.5 mm³, voxel size: 3.45 × 3.45 × 10 μm³) were collected as a *xyt* stack and converted into a *xyz* stack according to the guide provided on the Github site. The panels were prepared with NIS-Elements.

All mice were caged at a temperature of 23 ± 1°C, humidity of 50 ± 10%, and a 12-h light/dark cycle, with food and water ad libitum.

### descSPIM installed in the Japanese Foundation for Cancer Research

For Fig. 5a, the system installed in the Cancer Chemotherapy Center, Japanese Foundation for Cancer Research (JFCR) was built according to the guide provided on the descSPIM Github (Vortran, Stradus 405 nm/20 mW, Stradus 488 nm/25 mW, Stradus 640 nm/30 mW and Cobolt, Mambo 594 nm/50 mW; Laser light beam combiner: PNEUM, BeamCombiner Type-S). For Fig. 5d, G-292 clone A141B1 (JCRB cell bank #IFO50107) osteosarcoma cells stably expressing Akaluc and ZsGreen genes (G-292/Aka-ZsG) was generated by the lentivirus infection of pHIV-AKALuc-ZsGreen[66]. The plasmid pHIV-AKALuc-ZsGreen was created using the pHIV-Luc-ZsGreen (Addgene, #39196) and pcDNA3 Venus-Akaluc (provided by the RIKEN BRC through the National BioResource Project of the MEXT/AMED, Japan), and used for the lentivirus production in 293FT cells. Virus production, collection and infection were completed following the manufacturer's protocol. G-292/Aka-ZsG cells (1 × 10⁶ cells/mouse) were intravenously inoculated into 6-week-old female SCID-beige mice (Charles River). After six weeks of tumor inoculation, tumor-bearing mice were sacrificed with isoflurane and transcardially perfused with heparinized PBS and Mildform (Fujifilm #133-10311). Excised tissue samples were postfixed for 8–24 h and processed following the CUBIC tissue clearing protocols with some modifications. For Red-Dot2 (Biotium Inc #40061) staining, CUBIC-L-treated samples were immersed in CUBIC-HV2 staining buffer (CUBICStars #CSSR003) with 1 × RedDot2 at 37 °C for 2-3 days. The z-stack tiff data (approximately 7.4 × 10.7 × 5.0 mm³, voxel size: 3.45 × 3.45 × 10 μm³) were collected as a *xyt* stack and converted into a *xyz* stack according to the guide provided on the Github site. The panels were prepared with NIS-Elements.

Mice were housed in a specific pathogen-free room at 24 ± 2 °C and 50 ± 10% humidity, with a 12-h light/dark cycle. All mice were treated, monitored, and sacrificed following the protocol approved by the Committee for the Use and Care of Experimental Animals of JFCR in accordance with relevant guidelines and regulations. The maximum tumor volume in mice permitted by the committee is 2000 mm³ and the maximum tumor size was not exceeded in this study.

### descSPIM installed in the University of Tokyo

For Fig. 5a, the system installed in the University of Tokyo was built according to the guide provided on the descSPIM Github with some modifications (equipped three-color laser light source: Vortran, Stradus 488 nm/25 mW, 514 nm/60 mW, and 640 nm/60 mW; Laser light beam combiner: PNEUM, BeamCombiner Type-S). For Fig. 5e, the stomach from Chat-Cre; R26-LSL-TdTomato; Dclk1-Zsgreen mouse strains (JAX#06410, #07914[67]) (male) were cleared with CUBIC-L for 5 days at 37 °C, followed by CUBIC-R. The z-stack tiff data (approximately 7.0 × 7.9 × 5.2 mm³, voxel size: 3.45 × 3.45 × 10 μm³) were collected as a *xyt* stack and converted into a *xyz* stack according to the guide provided on the Github site. The panels were prepared with ImageJ.

Mice were housed in a specific pathogen-free, temperature-controlled room at 23 ± 1 °C and 50 ± 10% humidity with a 12-h light/dark cycle.

### descSPIM installed in Central Research Institute of Electric Power Industry

For Fig. 5a, the system installed in Central Research Institute of Electric Power Industry (CRIEPI) was built according to the guide provided on the descSPIM Github with some modifications (laser light source: Coherent OBIS LS 488 nm laser). For Fig. 5f, A male C57BL/6 mouse small intestine was delipidated with CUBIC-L (TCI, #T3740) for 2 days at 37 °C, followed by the whole-mount staining with propidium iodide (Sigma, #P4170) according to the manual of CUBIC-L (10 μg/mL in 0.1 M pH 7.4 Phosphate buffer containing 0.5 M NaCl). The sample was then RI-matched with CUBIC-R + (M) (TCI, #T3741). The z-stack tiff data (approximately 5.2 × 3.5 × 2.4 mm³, voxel size: 3.45 × 3.45 × 5 μm³) were collected as a *xyt* stack and converted into a *xyz* stack according to the guide provided on the Github site. The panels were prepared with NIS-Elements.

Mice were housed in a specific pathogen-free, climate-controlled room (temperature 23 ± 2 °C, humidity 50 ± 20%) with a 12-h light/dark cycle.

### descSPIM installed in Nippon Medical School

For Fig. 5a, the system installed in Nippon Medical School was built according to the guide provided on the descSPIM Github. For Fig. 5g, *Tg(fli1a:myr-EGFP)^{ncv2Tg}* zebrafish line, in which endothelial cells were labeled with EGFP fluorescence[68], was generated using the Tol2 transposon system[69]. *Tg(fli1a:myr-EGFP)^{ncv2Tg}* adult zebrafish (approximately 1 cm in body length) was delipidated with CUBIC-L (TCI) for 1 day at 37 °C. The sample was then RI-matched with CUBIC-R (TCI). The z-stack tiff data (approximately 14.1 × 6.2 × 2.9 mm³, voxel size: 3.45 × 3.45 × 3.3 μm³) were collected as a *xyt* stack and converted into a *xyz* stack according to the guide provided on the Github site. The panels were prepared with Fiji software.

Zebrafish were grown and maintained on a 14-h/10-h light/dark cycle at 28°C.

### descSPIM installed in Kyushu University

For Fig. 5a, the system installed in Kyushu University was built according to the guide provided on the descSPIM Github with some modifications (tube lens: Thorlabs, TTL200-A; objective lenses: Nikon, CFI Plan Fluor DL 10XF CH, laser light source: Vortran, Stardus 488-25). For Fig. 5g, pregnant mice were sacrificed on E7.5 and their uteri were harvested and slit open to expose each decidua. The decidua were collected in 1 mL of 4% PFA overnight and were delipidated with CUBIC-L (TCI, T3740) for 1 day at 37 °C, followed by the whole-mount staining with SYTOX-Green (Thermo Fisher Scientific, S7020). The sample was then RI-matched with CUBIC-R + (M) (TCI, T3741). The z-stack tiff data (approximately 1413 × 745 × 750 μm³, voxel size: 0.345 × 0.345 × 4 μm³) were collected as a *xyt* stack and converted into a *xyz* stack according to the guide provided on the Github site. The panels were prepared with ImageJ.

Mice were housed in a specific pathogen-free, temperature-controlled room at 22 °C, and 55 ± 10% humidity, with a 12-h light/dark cycle.

### descSPIM installed in Kyoto Prefectural University of Medicine

For Fig. 5a, the system installed at Kyoto Prefectural University of Medicine was constructed according to the instructions provided on the descSPIM Github (equipped three-color laser light source: Vortran, Stradus 488 nm/25 mW, 561 nm/50 mW, and 640 nm/30 mW; Laser light beam combiner: PNEUM, BeamCombiner; objective lens: Olympus RMS4X). For Fig. 5i, a *RARE-lacZ* Transgenic (Tg) mouse embryo (E12.0)[70] (provided by RIKEN BioResource Research Center,

#RBRC06571) was delipidated with CUBIC-L for 2 days at 37 °C. For nuclear staining, the sample was incubated overnight at 37 °C in Sca-*l*eCUBIC-1A with 500 mM NaCl using SYTOX-Green (Thermo Fisher Scientific, #S7020) at a dilution of 1/2500. Prior to whole mount staining with antibodies, primary antibodies were first preincubated with fluorescence-conjugated Fab fragments specific to Fc fragment; goat anti-chicken IgY Fab fragment Alexa 647 conjugate (Jackson, # 103-607-008) for anti-β-galactosidase antibody (Abcam, #ab9361) and goat anti-rabbit IgG Fab fragment Cy3 conjugate (Jackson, #111-167-008) for anti-cleaved Caspase3 antibody (Cell Signaling Technology, #9661) at 32 °C with a ratio of primary antibody to secondary Fab fragment of 1:0.7. The sample was then incubated in a 1/200 primary antibody concentration in HEPES-TSC buffer at 32 °C for 4 days, followed by post fixation in 1% formaldehyde in 0.1 M PB for 1 h at room temperature. Subsequently, the sample was RI-matched using CUBIC-R. The *z*-stack tiff data (approximately $6.35 \times 3.35 \times 5.76$ mm$^3$, voxel size: $1.55 \times 1.55 \times 8.59$ μm$^3$) were collected as a *xyt* stack and converted into a *xyz* stack according to the guide provided on the Github site. The panels were prepared with NIS-elements.

All mice were maintained under controlled conditions (12-h light/dark cycle, temperature of 24 °C) and were provided with free access to water and a standard pellet diet.

### descSPIM installed in University of Occupational and Environmental Health
For Fig. 5a, the system installed in University of Occupational and Environmental Health was built according to the guide provided on the descSPIM Github with some modifications (equipped four-color laser light source: Vortran, Stradus, 405 nm/25 mW, 488 nm/25 mW, 561 nm/50 mW, and 640 nm/30 mW; Laser light beam combiner: PNEUM, BeamCombiner Type-S).

### descSPIM installed in Niigata University, the University of Nebraska-Lincoln, Tokyo Medical and Dental University, National Cancer Research Center, Tsukuba University, Hokkaido University, and Chiba University
For Fig. 5a, the systems installed in Niigata University, the University of Nebraska-Lincoln, Tokyo Medical and Dental University, National Cancer Research Center, Tsukuba University, Hokkaido University, and Chiba University were built according to the guide provided on the descSPIM Github.

### Statistics and reproducibility
For the application examples (Figs. 2–5), no statistical method was used to predetermine sample size, no data were excluded from the analyses, the experiments were not randomized, and the investigators were not blinded to allocation during experiments and outcome assessment. These choices were made because these were demo/pilot samples and testing specific biological hypotheses required additional experiments. The spatial resolution measurements (Figs. 1, 2, Supplementary Fig. 4, and Supplementary Fig. 12) have inherently negligible inter-sample variability because they use fixed calibration samples (fluorescent beads and quantum dots).

### Reporting summary
Further information on research design is available in the Nature Portfolio Reporting Summary linked to this article.

## Data availability
The imaging data obtained in this study are available upon request to the corresponding author (suishess-kyu@umin.ac.jp). Raw data for CDX analysis and whole mouse brain reconstruction are too large to be provided with this paper. Requests for data accessibility will be fulfilled with 4 weeks. Source data to make graphs are provided with this paper. Source data are provided with this paper.

## Code availability
Our custom codes for the descSPIM control software, the flat-field correction, tiling light-sheet method, ANTs registration, compiling NIfTI files from the TIFF stack, and quantification of drug distribution are provided on our GitHub website (https://github.com/dbsb-juntendo/descSPIM). The minimum dataset for test running the original code used in this paper can be found at our GitHub website (https://github.com/dbsb-juntendo/descSPIM).

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

## Acknowledgements

The authors thank the lab members at DBSB Juntendo, particularly Y. Wada for setting data analysis environments, S. Yanagi and D. Inahara for helping 3D CAD preparation. We also thank the Biomedical Research Core Facilities, Juntendo University Graduate School of Medicine, Robert T. Huang Entrepreneurship Center of Kyushu University for technical assistance, M. Suzuki (National Cancer Center Research Institute) for assisting CDX sample preparation, A. Yoshikawa (Nagasaki Univ. and Kameda Medical Center) for the pathological evaluation of Fluo-HE CDX image, and T. Kanai (CRIEPI) for sharing the CMOS camera to construct the system in CRIEPI. This study was supported by the Japan Agency for Medical Research and Development (AMED) PRIME (to E.A.S., and Y.Hayakawa, grant number JP20gm6210027 and JP23gm6510018); AMED Project for Cancer Research and Therapeutic Evolution (P-CRE-ATE) (to S.T., grant number JP21cm0106286); AMED Research on Development of New Drugs (to A.H. and E.A.S., grant number JP21ak0101181); AMED Brain/MINDS (to E.A.S., K.T., and T.Nemoto, grant number JP21wm0425003, JP21wm0425001, and JP19dm0207078); AMED Moonshot program (to K.T., grant number JP21zf0127004); AMED SCARDA (to K.N., grant number JP223fa827004); AMED Project for Regenerative/Cellular Medicine and Gene Therapies (to N.S. and K.K., grant number JP23bm1423012 and JP23bm1123032); AMED-CREST (to N.S., grant number JP21gm1410009); AMED Project for Promotion of Cancer Research and Therapeutic Evolution (P-PROMOTE) (to S.T. and E.A.S., grant number JP23ama221220 and JP22ama221517); AMED The iD3 Booster (to J.K., grant number JP23nk0101665); Japan Science and Technology Agency (JST) CREST (to K.Otomo and E.A.S., grant number JPMJCR20E4 and JPMJCR23B7); JST Adaptable and Seamless Technology transfer Program through Target-driven R&D (A-STEP) (to T.Nii, grant number JPMJTR22UA); JST Moonshot program (to I.M. grant number JPMJMS2023); Japan Society for the Promotion of Science (JSPS) KAKENHI grant-in-aid for scientific research (B) (to K.Otomo, K.N., K.Otsuka, S.F., K.Y., M.K., K.Sekine, F.K., and E.A.S., grant number JP22H02756, JP22H02523, JP19H04274, JP21H02665, JP21H03127, JP23H02878, JP22H03141, JP20H03549, and JP22H02824); JSPS KAKENHI grant-in-aid for scientific research (S) (to T.Nemoto and K.Otomo, grant number JP20H05669); JSPS KAKENHI grant-in-aid for challenging research (exploratory) (to K.Otomo, K.N., M.K. and S.F., grant number JP21K19346, JP23K18081, JP21K19895, and JP21K19358); JSPS KAKENHI grant-in-aid for scientific research (C) (to N.S. and T.Nii, grant number JP19K06896 and JP22K06810); JSPS KAKENHI grant-in-aid for research activity start-up (to R.Y. grant number JP21K20703); JSPS KAKENHI grant-in-aid for challenging research (pioneering) (to R.K. grant number JP22K183838); JSPS KAKENHI grant-in-aid for scientific researches on innovative areas "Adaptation Circuit Census" (to F.K., grant number JP21H05241); JSPS KAKENHI grant-in-aid for transformative research areas - platforms for advanced technologies and research resources "Advanced Bioimaging Support" (to T.Nemoto and E.A.S., grant number JP22H04926); JSPS KAKENHI grant-in-aid for fund for the promotion of joint international research (Fostering Joint International Research(B)) (to K.Otomo, grant number JP22KK0100); JSPS KAKENHI for international leading research (to E.A.S., grant number JP23K20044); National Institutes of Health (NIH) COBRE (to T.Y. through the Nebraska Center for the Prevention of Obesity Diseases, grant number GM104320); Operating Costs Subsidies for Private Universities (to E.A.S.); Grants-in-Aid from UTEC-UTokyo (to E.A.S.); the Takeda Science Foundation (to E.A.S.); Nakatani foundation for advancement of measuring technologies in biomedical engineering (to E.A.S.); The Mochida Memorial Foundation for Medical and Pharmaceutical Research (to E.A.S.), The Uehara Memorial Foundation (to K.N. and E.A.S.), Nebraska Tobacco Settlement Biomedical Research Development Funds (to T.Y.), National Cancer Center Research and Development Fund (to K.Sekine, 2021-A-02) and a grant of the Research Grant of the Princess Takamatsu Cancer Research Fund (to K.Sekine, 21-25322).

## Author contributions

E.A.S. and K.Otomo designed the project and constructed the microscope. K.Otomo, Y.N., S.J.E., Y.Sato, H.B., and E.A.S. developed the 3D imaging procedures. Y.Saito, H.U., K.T., and E.A.S. prepared the cleared and stained samples. Y.W. and T.Nemoto provided the brain samples of *Thy1*-YFP-H and *Thy1*-GFP-M Tg mice. S.Y. and A.H. provided the CDX samples. K.Otomo acquired imaging data. T.O. and E.A.S. wrote the codes for image processing. T.O. performed image registration and fusion with ANTs and Bigstitcher. T.O. also developed the algorithm for the quantitative analysis of drug distribution. Y.N. and T.O. prepared the descSPIM GitHub website. K.N., R.Y., N.S., S.T., R.K., Y.I., T.S., Y.Hayakawa, K.Ostuka, H.W.T., Y.Haneda, S.F., M.F. T.Nii, C.M., N.T., K.Y., J.M.R.R., M.K., T.Y., Y.O., H.Koike, Y.C., K.Sekine, J.K., K.Sugiyama, K.K., F.K., H.Kim, and I.M. constructed descSPIM on their site and provided the photograph of their system. Moreover, K.N., R.Y., N.S., S.T., R.K., Y.I., T.S., Y.Hayakawa, K.Ostuka, H.W.T., Y.Haneda, S.F., M.F. T.Nii, C.M., N.T., and K.Y. provided acquired 3D image data. K.Otomo, T.O., and E.A.S. wrote the manuscript with input from all co-authors.

## Competing interests

E.A.S. and K.T. are co-inventors on patents and patent applications owned by RIKEN covering the CUBIC reagents, and E.A.S. is employed by CUBICStars Inc. that offers services based on CUBIC technology. The remaining authors declare no competing interests.

## Additional information

[1]Department of Biochemistry and Systems Biomedicine, Juntendo University Graduate School of Medicine, Tokyo, Japan. [2]Biochemistry II, Juntendo University School of Medicine, Tokyo, Japan. [3]Nakatani Biomedical Spatialomics Hub, Juntendo University Graduate School of Medicine, Tokyo, Japan. [4]Division of Biophotonics, National Institute for Physiological Sciences, National Institutes of Natural Sciences, Okazaki, Japan. [5]Biophotonics Research Group, Exploratory Research Center on Life and Living Systems, National Institutes of Natural Sciences, Okazaki, Japan. [6]Department of Neurosurgery, University of Tokyo, Tokyo, Japan. [7]Department of Neurosurgery and Neuro-Oncology, National Cancer Center Hospital, Tokyo, Japan. [8]Science for Life Laboratory, Department of Applied Physics, KTH Royal Institute of Technology, Stockholm, Sweden. [9]Department of Pharmacology and Therapeutics, Fundamental Innovative Oncology Core, National Cancer Center Research Institute, Tokyo, Japan. [10]Division of Molecular Pharmacology, National Cancer Center Research Institute, Tokyo, Japan. [11]Department of System Pathology for Neurological Disorders, Brain Research Institute, Niigata University, Niigata, Japan. [12]Department of Basic Veterinary Science, Joint Faculty of Veterinary Medicine, Kagoshima University, Kagoshima, Japan. [13]Advanced Neuroimaging Center, Institute for Quantum Medical Sciences, National Institutes for Quantum Science and Technology, Chiba, Japan. [14]Division of Experimental Chemotherapy, Cancer Chemotherapy Center, Japanese Foundation for Cancer Research, Tokyo, Japan. [15]Department of Gastroenterology, Graduate School of Medicine, The University of Tokyo, Tokyo, Japan. [16]Biology and Environmental Chemistry Division, Sustainable System Research Laboratory, Central Research Institute of Electric Power Industry, Chiba, Japan. [17]Department of Molecular Pathophysiology, Institute for Advanced Medical Sciences, Nippon Medical School, Tokyo, Japan. [18]Department of Developmental Biology, Graduate School of Medical Sciences, Kyushu University, Fukuoka, Japan. [19]Anatomy and Developmental Biology, Kyoto Prefectural University of Medicine, Kyoto, Japan. [20]Department of Pediatrics, Kyoto Prefectural University of Medicine, Kyoto, Japan. [21]Division of Bio-prosthodontics, Faculty of Dentistry & Graduate School of Medical and Dental Sciences, Niigata University, Niigata, Japan. [22]Department of Biochemistry, University of Nebraska-Lincoln, Lincoln, USA. [23]Department of Meidical Biochemistry, Tokyo Medical and Dental University, Tokyo, Japan. [24]Laboratory of Cancer Cell Systems, National Cancer Center Research Institute, Tokyo, Japan. [25]The Second Department of Internal Medicine, University of Occupational and Environmental Health, Kitakyushu, Japan. [26]Institute for Advanced Research of Biosystem Dynamics, Research Institute for Science and Engineering, Waseda University, Tokyo, Japan. [27]Life Science Center for Survival Dynamics, Tsukuba Advanced Research Alliance (TARA), University of Tsukuba, Tsukuba, Japan. [28]Lab of Histology and Cytology, Graduate School of Medicine, Hokkaido University, Sapporo, Japan. [29]Department of Systems Medicine, Graduate School of Medicine, Chiba University, Chiba, Japan. [30]These authors contributed equally: Kohei Otomo, Takaki Omura.
✉e-mail: suishess-kyu@umin.ac.jp

