## [Peer Review File · Nature Communications]

Reviewers' Comments:

Reviewer #1:

Remarks to the Author:

The manuscript by Kohei Otomo et al. presents the development of an affordable, easy-to-build light-sheet microscope for examining cleared tissue samples. The authors have named their microscope 'descSPIM' and state that it can be built for only \$20,000–50,000 in one day by a non-expert. They demonstrate the capability of the microscope by examining a PI-stained mouse brain hemisphere and a whole Thy1-YFP-H mouse brain, as well as a tumor mass from a cancer cell line-derived xenograft. The manuscript is well-written and describes in detail how to build the descSPIM microscope. The figures are illustrative, and the supplementary videos are visually compelling.

This paper represents a valuable contribution to the field of microscopy, particularly by highlighting a cost-effective design. The emphasis on affordability is commendable and addresses a significant barrier in accessing advanced imaging technology. However, the paper does not sufficiently highlight the technical innovations inherent in the microscope's development. Light-sheet microscopy is a rapidly evolving field, and each new design typically incorporates novel technical aspects or improvements over existing systems. The work described by Otomo and colleagues undoubtedly includes such innovations and unique design features. Elaborating on the unique features of the microscope design compared to existing light-sheet microscopes, discussing any novel technical solutions employed to overcome specific challenges in light-sheet microscopy, and providing a more detailed analysis of how these innovations contribute to the system's performance and capabilities, especially in contexts where standard commercial systems might fall short, would significantly enhance the paper's impact and scientific value.

The control software is critical to any automated or semi-automated microscopy system. The authors note that "The basic system does not require users to deploy any custom programs for operations; it can be used only with the device-associated software." Including a description of the software modules used to control the microscope, especially any custom-developed components, would be beneficial. The authors should also consider developing a control software that can manage all the hardware and image acquisition. The descSPIM would significantly benefit from having such a graphical interface for users.

The authors state that nearly 40 users actively utilize the descSPIM technology, demonstrating the practicality and appeal of the design. However, the manuscript would greatly benefit from including data or findings from some of these users. Adding data, or at least a table, from various users would provide empirical evidence of the microscope's capabilities and performance in diverse settings, as well as showcase its versatility and adaptability in real-world applications. Data from different users can serve as robust validation of the microscope's performance across a range of experiments and conditions. Insights into how users apply the microscope in their specific research areas would offer valuable information on its ease of use, adaptability, and effectiveness in various scientific contexts. Presenting a variety of applications from different users could highlight the versatility of the descSPIM microscope, potentially broadening its appeal to a wider audience. User data can also provide feedback that may be useful for further refinement and development of the microscope.

Consider revising the title from "descSPIM: affordable and easy-to-build light-sheet microscopy for tissue clearing technique users" to "descSPIM: An Affordable and Easy-to-Build Light-Sheet Microscope Optimized for Tissue Clearing Techniques".

Reviewer #2:

Remarks to the Author:

In this manuscript, Otomo and colleagues describe the design and implementation of a simple, easy to build and relatively inexpensive light sheet microscope with fully open-sourced documentation and support. First, I want to applaud the authors for the quality of both the assembly instructions in their GitHub page as well as the high-end quantifications of the optical capabilities. The title of the manuscript fits perfectly with what the descSPIM can offer: an

"affordable and easy-to-build light-sheet microscopy for tissue clearing technique users". From my reading of the manuscript, one of the main goals is to make the system as inexpensive as possible but still practical to use. Unfortunately, in the pursuit of cutting costs, some of the set-up configurations make the system not very practical for a non-microscopy and non-image analysis specialists. Please find my recommendations and concerns below:

1. While the title of the paper describe perfectly what descSPIM can offer, the rest of the manuscript is often over selling or too ambitious in the application range offered. I think descSPIM could be extremely useful to answer questions such as "where are my cells located within my organ?". This will advance fields such as immuno-oncology (where are my immune cells compared to my tumor? How are tumor cells spread within my organ?...." The system offers barely cellular resolution, more likely cell population resolution. And it is OK, a lot of scientific questions require low resolution so it will be extremely useful. For instance, in line 227, authors infer, based on their image that they reach cellular to subcellular resolution. It is more than a stretch. Overall, I think the paper will benefit from a better discussion about not only what the system can offer but more importantly what are the limitations of the system (which a will describe in more details below). In their discussion the authors propose to upgrade the system with add-ons to remove the limitations but then the system will become a lot more expensive and a lot more challenging to maintain by non-microscopist experts. A comparison with the Benchtop mesoSPIM (Vladimirov et al., 2023). I believe they have a niche within the simple and affordable landscape and should stay there with a better description of the limitations of the system. It will be a very useful microscope.
2. The choice of non-motorized x, y and O stages makes precise tiling for large samples a challenge not only during acquisition but most importantly for image registration, tiling and fusion. In my opinion. Image processing is a bigger challenge for end users than optics or cost which render the system not as practical. A motorized stage and an operating software such as micro-manager will allow to write the required metadata in the TIFFs file making the image processing a lot easier for non-experts. This choice also begs the question of how reproducible image acquisition can be? The manual rotation of a sample in a cuvette for instance can end in sample displacement making registration challenging. Could the authors, if they don't have quantified data on reproducibility, at least discussed this limitation in the discussion section?
3. The choice of having a single illumination side is understandable to limit cost but having to take images at 2 angles can significantly increase the file size which now place the financial burden in data storage and processing. Could the authors provide and discussed data size for their examples?
4. The comparison between CUBIC and BABB treated samples is important and unfortunately only purely qualitative. Could the authors provide at least PSF measurements and light sheet dispersal quantification in various RIMS broadly used? (Maybe adding Ethyl Cinnamate?).
5. In most of the qualitative figures, such as Fig3, the authors only zoom in on the side of the tissue where the illumination and detection will be optimal. Can they provide images of the inside of the tissue to visualize the loss of resolution across?
6. L227: "overall quality was comparable to our advanced light-sheet microscopy system". This is again, a stretch. If no quantification, could the authors provide side by side image comparisons of the 2 system at the same magnification in supp Fig11?
7. The usefulness of the descSPIM for pathologist should be either removed or at least toned down. As the authors state, they obtained "relatively low resolution for usual pathology examination". L274 should be removed since it is not a scientific argument "a pathologist concurred...".

Details and typo/phrasing:

1. L61: I think the authors mean "adapted" instead of "adopted"
2. Replace several instances of "obtained" with something like "generated" for data they produced (L247 and 267).
3. SuppFig1: on the mesoSPIM line I think the authors mean "Facility" instead of "Faculty".

Reviewer #3:

Remarks to the Author:

This article entitled: "descSPIM, affordable and easy-to-build light-sheet microscopy for tissue clearing technique users" present a simple but efficient light sheet mmicroscope specificaaly for users of clearing techniques.

It is a welcome new addition to the already long list of options in the DIY world of Light Sheet. It is supposed to fill a gap as most companies in this sector have all spent vast amount of money on expensive systems to target the users of the ever expanding users of clearing techniques. Unfortunately most of those users need to have access to an imaging platform or a well developed institute as those scopes are expensive. The descSPIM, a word play with desk and SPIM more than a real acronym, is well documented and well presented but I have several concerns as it requires some modifications in order to be more readable and more accessible to its target audience. I will recommend some major and minor revisions to improve it and make it better for the community and its users.

Major revisions:

1. The descSPIM is presented as an open science tool and it is disappointing to see that the community aspect is in the discussion. This is a result as it is rare to get a new tool which has already been used and developed outside its original lab, with several workshops. It will be important to relocate this element in the results section with maybe a figure showing a map of the already expanding new descSPIMs and pictures from the workshop. It is essential that new tools are not single machines in a lab but already replicated before to be published and this is there so please show it.

2. the introduction of LSFm is a bit simplistic and only discussed the SPIM and the DSLM and nothing on the single objective, the mirror based approach and so on so it will benefit from a better presentation as it is currently confusing as diSPIM is presented much later. The DSLM is not even properly referenced!!!

3. the use of "basic" in the results does not make sense as the only advanced part concerned processing and not the system itself. Moreover Figure 1 is nice but needs some work as first the panel a does not define what "advanced" means to the authors? the cost could be numbered as the price of the descSPIM is provided as well as the 2 mesoSPIM options and it will be good to also mention the desktop version of the mesoSPIM which is already published rather than focused on the most expensive one!

Panel b is fine but the colour coding is not reused in the next panels. Panel c and d showing perspectives of a black block without much shadow is not helping. Maybe a better choice of colour and shading including transparency should improve the reader read-out of the scope structure and please check your spelling: Excitation not Excitation (panel c). And using the exact same angle of perspective without overlapping labels on the design will help too! Then in panels e to g this is an overwhelming amount of data per inch square. Why do you need to spell out the FWHM already explained in the text and visible in the panel above, you just repeat the information and the figure will be so small that it makes no sense, please do improve this!

4. acronym bonanza: you use an acronym in the abstract but it is undefined and along the text many acronyms paved the way to SAOD (severe acronyms disorders). It is hard in those 2 fields to avoid using them but then use them and define them as CUBIC and others are just referenced so many could get the same faith!

5. title: it seems that using microscopy is not correct as you are presenting not a technique but an instrument so it should be microscope...

Minor revisions:

6. In results section 1 you mention device associated software? Could not find any mention of it in the method? As well later you state minute-order volumetric imaging per stack, any value?

7. lines 216 to 218: you explain about fusion issues and say they are "globally" rather vague, isn't it. It is probably due to the limited number of views you use and the limited overlap as well as the detection decay of the signal.

8. equations lines 466 and 468.. it should be spelled dZsample and not dZsample

9. line 525 please show some respect to other open science projects and please reference Fiji not

only a mention, not cool!

I do hope this long list of revisions is not scaring the authors and the editors as this paper should get published as not only users needs it but also maybe developing countries where LSFMF is unfortunately a luxury.

Fingers crossed,

Dr Emmanuel G.Reynaud, AKA the acronym Guru

Dear Reviewers

Thank you for your comments on our manuscript, of which title was changed in accordance with reviewer #1's recommendation: descSPIM: An Affordable and Easy-to-Build Light-Sheet Microscope Optimized for Tissue Clearing Techniques. We are pleased to note favorable comments and helpful suggestions from the reviewers. We have carefully studied the comments and made revisions to the manuscript.

During the revision process, we identified a point that should be corrected in *Supplemental Fig. 5* and we have additionally included the following statement to *Results*.

Results (P 10, L 216–224) now reads:

However, the 830 μm measurement is narrower than the estimated effective field of view (eFOV) in the clearing reagent based on the measured beam profiles in the air (*Supplementary Fig. 5*). The higher RI elongates the Rayleigh length by a factor equal to the RI value compared to its length in the air. The pixel size of our utilized beam profiler was $4.4 \times 4.4 \mu\text{m}^2$, which is comparable to the estimated axial FWHM of the light intensity profile. This similarity might result in an overestimation of the light intensity profile. The eFOV calculated in FF mode at NA 0.007 is 16 mm in an environment with a refractive index of 1.52, which aligns with the measured value of 11 mm in the air (*Supplementary Fig. 5*). With these considerations, the cylindrical lens used in FA mode yields a sheet illumination near to the approximation.

Dear Reviewer #1

We would like to thank the reviewer #1 for the important comments, which enabled us to improve the quality of our manuscript. Here are our responses to your insightful comments.

Comment #1

This paper represents a valuable contribution to the field of microscopy, particularly by highlighting a cost-effective design. The emphasis on affordability is commendable and addresses a significant barrier in accessing advanced imaging technology. However, the paper does not sufficiently highlight the technical innovations inherent in the microscope's development. Light-sheet microscopy is a rapidly evolving field, and each new design typically incorporates novel technical aspects or improvements over existing systems. The work described by Otomo and colleagues undoubtedly includes such innovations and unique design features. Elaborating on the unique features of the microscope design compared to existing light-sheet microscopes, discussing any novel technical solutions employed to overcome specific challenges in light-sheet microscopy, and providing a more detailed analysis of how these innovations contribute to the system's performance and capabilities, especially in contexts where standard commercial systems might fall short, would significantly enhance the paper's impact and scientific value.

Authors' response:

We thank the kind suggestion from Reviewer #1. According to your kind encouragement, we added descriptions about the technical innovations and unique design features of descSPIM in *Discussions*;

Discussions (P 18-19, L 432-454) now reads:

The descSPIM, despite its simplistic design and configuration, boasts remarkable versatility suitable for a broad spectrum of imaging applications essential for studying cleared tissues. To achieve this, descSPIM incorporates improvements in both new technical aspects and existing systems, incorporating unique design features. In comparison to existing microscopes, descSPIM showcases pioneering technical solutions addressing specific challenges encountered in LSFM. Firstly, by eliminating the medium chamber and employing a two-stage synchronization approach, descSPIM enables 3D imaging, enhancing handling ease and accommodating various sample sizes and clearing reagents. Additionally, the streamlined process of capturing xyt images and reconstructing them into xyz images significantly reduces imaging time, allowing for z-stack image acquisition within a few minutes, a notable improvement over traditional methods such as our advanced LSFM system, Gemini (39), which typically took 15 minutes. Notably, the FF mode allowed for the efficient imaging of a whole mouse brain hemisphere in just minutes, ensuring a one-step completion of the process (**Fig. 2a and Supplementary Fig. 8**). Switching to FA mode and tiling LSFM method achieved a spatial resolution of 4 μm in the lateral direction and 7 μm in the axial direction, allowing for nearly isotropic imaging of labeled nuclei (**Fig. 2c**). Moreover, the advanced implementations of the device for biomedical purposes, in conjunction with the full utilization of image processing techniques, demonstrated the successful data acquisition, visualization,

and quantitative analysis of whole volumetric samples of the order of cm^3 (**Figs. 3, 4 and Supplementary Fig. 12**). Remarkably, descSPIM's compact design, utilizing a minimal number of commercially available optical elements on a modest-sized breadboard, renders it suitable for personal use and particularly valuable in regions where access to high-end systems is limited.

Comment #2

The control software is critical to any automated or semi-automated microscopy system. The authors note that “The basic system does not require users to deploy any custom programs for operations; it can be used only with the device-associated software.” Including a description of the software modules used to control the microscope, especially any custom-developed components, would be beneficial. The authors should also consider developing a control software that can manage all the hardware and image acquisition. The descSPIM would significantly benefit from having such a graphical interface for users.

Authors' response:

We appreciate the striking comment from Reviewer #1. For more precise usage, we installed a low cost USB to IO expansion board and synchronized the initiation of camera acquisition with stage movements to automate z-stack acquisition. The camera is controlled in μ Manager and the stages with napari-micromanager widget. We compared reproducibilities of z-stack acquisition between this method and the original method using two-kind of device-associated software, realizing that both methods have inherent advantages. For users to select both methods, we added descriptions about this automation of acquisitions in **Results, Discussions, Material and Methods**, and **Supplementary Figs. 9-11**. and uploaded detailed procedures in our GitHub website;

Results (P 12, L 262-271) now reads:

We validated the reproducibility of z-stack measurements and identified axial shifts in the stack resulting from disparities in start timing (**Supplementary Fig. 9a**). However, these discrepancies were effectively mitigated through adjustments solely in the z-direction (**Supplementary Fig. 9b**). Notably, no shifts were detected in the xy-plane, ensuring consistent acquisition reproducibility (**Supplementary Fig. 9a**). To address axial displacements, we provided custom software allowing users to specify start timing and ensure z-position (**Supplementary Fig. 10**). Similar to **Supplementary Fig. 9**, this acquisition method was verified to enable reproducible image acquisition (**Supplementary Fig. 11**). Indeed, it effectively reduced axial displacement, rendering it an ideal option for experiments requiring precise position reproducibility (**Supplementary Fig. 11**).

Discussions (P 19-20, L 455-463) now reads:

For controlling this minimal LSFM, we initially proposed a method of simultaneously driving two different device-associated software. While this procedure enabled reproducible image acquisition, axial displacement for a few slides might occur. The required manual synchronization made it difficult to achieve complete elimination of human error, and in this case, post-correction of displacement of a few slides is easy. To automate

the z-stack acquisition, a low-cost USB to IO expansion board was employed to synchronize camera initiation with stage movement. The latter, the custom software-based approach excels in reproducibility, but due to the setup complexity, we recommend prioritizing the utilization of the former, device-associated software for its simplicity.

Materials and Methods (P 26, L 627-631) now reads:

In addition, a low-cost USB to IO expansion board (Arduino Uno Rev3) was employed to synchronize camera initiation with stage movement to automate z-stack acquisition (**Supplementary Fig. 10**). A simple Jupyter Notebook was developed to guide users through a z-stack acquisition (available on GitHub). The camera was controlled in μ Manager, and the stages were controlled with a napari-micromanager widget.

Comment #3

The authors state that nearly 40 users actively utilize the descSPIM technology, demonstrating the practicality and appeal of the design. However, the manuscript would greatly benefit from including data or findings from some of these users. Adding data, or at least a table, from various users would provide empirical evidence of the microscope's capabilities and performance in diverse settings, as well as showcase its versatility and adaptability in real-world applications. Data from different users can serve as robust validation of the microscope's performance across a range of experiments and conditions. Insights into how users apply the microscope in their specific research areas would offer valuable information on its ease of use, adaptability, and effectiveness in various scientific contexts. Presenting a variety of applications from different users could highlight the versatility of the descSPIM microscope, potentially broadening its appeal to a wider audience. User data can also provide feedback that may be useful for further refinement and development of the microscope.

Authors' response:

We surely thank this important recommendation from Reviewer #1. According to this, we contacted descSPIM end-users to provide photographs of their descSPIM and acquired 3D image data, realizing that descSPIM dissemination has steadily proceeded. About this great dissemination, we added them as authors, descriptions in **Results, Material and Methods** (P 35-42, L 840-1003) and **Fig. 5**;

Results (P 17 L 389-408) now reads:

Dissemination of descSPIM in the research community

To facilitate the dissemination of descSPIM, we have conducted annual tutorial courses on its construction for end-users interested in tissue clearing techniques. Drawing inspiration from previous open-source initiatives such as openSPIM (45), mesoSPIM (38, 40), diSPIM (29, 30), and legolish/lemolish (<https://lemolish.mystrikingly.com/>), as well as embracing the ethos of open microscopy (57), we have established a dedicated GitHub repository (<https://github.com/dbsb-juntendo/descSPIM>). This repository hosts comprehensive instructions for building descSPIM and fosters community engagement. As a result of these

efforts, nearly 40 research groups worldwide have adopted descSPIM or are considering to install, indicating its ability to address the growing demand for versatile tissue clearing applications (**Fig. 5a**). Indeed, descSPIM has been successfully applied in various biological contexts; imaging of a rat-brain coronal slice immunostained for tyrosine hydroxylase in the hypothalamus (**Fig. 5b**); visualization of a mouse brain hemisphere stained with SYTOX-Green (**Fig. 5c**); examination of ZsGreen-labelled tumor metastasis in mouse lung, stained with RedDot2 (**Fig. 5d**); observation of a stomach from Chat-Cre; R26-LSL-TdTomato; Dclk1-Zsgreen mouse strains (**Fig. 5e**); labeling of mouse intestine with SYTOX-Orange (**Fig. 5f**); imaging of *Tg(fli1a:myr-EGFP)^{ncv2Tg}* zebrafish expressing EGFP in endothelial cells (**Fig. 5g**); visualization of z-stack images of E7.5 decidua labelled with SYTOX-Green (**Fig. 5h**); and 3D imaging of E12.0 RARE-lacZ whole mouse embryo (**Fig. 5i**). These data indicate that descSPIM is an affordable and easy-to-build microscope as well as a fully useful microscope.

Comment #4

Consider revising the title from “descSPIM: affordable and easy-to-build light-sheet microscopy for tissue clearing technique users” to “descSPIM: An Affordable and Easy-to-Build Light-Sheet Microscope Optimized for Tissue Clearing Techniques”.

Authors' response:

We agreed insightful comment from Reviewer #1. According to your suggestion, we revised the title from “descSPIM: affordable and easy-to-build light-sheet microscopy for tissue clearing technique users” to “descSPIM: An Affordable and Easy-to-Build Light-Sheet Microscope Optimized for Tissue Clearing Techniques”.

Dear Reviewer #2

We are grateful for the complete agreement regarding the title. In the revised manuscript, though, we have changed the title to "descSPIM: An Affordable and Easy-to-Build Light-Sheet Microscope Optimized for Tissue Clearing Techniques," in accordance with reviewer #1's recommendation. We believe this revised title will likewise receive agreement.

Also, we would like to thank the reviewer #2 for the important comments that significantly helped improve the quality of our manuscript. Here are our responses to your insightful comments.

Comment #1

While the title of the paper describe perfectly what descSPIM can offer, the rest of the manuscript is often over selling or too ambitious in the application range offered. I think descSPIM could be extremely useful to answer questions such as “where are my cells located within my organ?”. This will advance fields such as immunology (where are my immune cells compared to my tumor? How are tumor cells spread within my organ?....” The system offers barely cellular resolution, more likely cell population resolution. And it is OK, a lot of scientific questions require low resolution so it will be extremely useful. For instance, in line 227, authors infer, based on their image that they reach cellular to subcellular resolution. It is more than a stretch.

Overall, I think the paper will benefit from a better discussion about not only what the system can offer but more importantly what are the limitations of the system (which a will describe in more details below). In their discussion the authors propose to upgrade the system with add-ons to remove the limitations but then the system will become a lot more expensive and a lot more challenging to maintain by non-microscopist experts. A comparison with the Benchtop mesoSPIM (Vladimirov et al., 2023). I believe they have a niche within the simple and affordable landscape and should stay there with a better description of the limitations of the system. It will be a very useful microscope.

Author's response:

We thank for the reviewer's helpful advice and support by addressing “descSPIM could be extremely useful to answer questions,” regarding low resolution imaging. The additional point that this reviewer mentioned was due to the ambiguous use of the word “subcellular” without any numerical definition. Our system's resolution was about 4 μm in lateral and 7.2 μm in axial with FA mode (expanded 1.5x by CUBIC, leading to a practical resolution of 2.7 μm in lateral and 4.8 μm in axial), which we consider justified its description as “subcellular”, literally the resolution comparable to the structure smaller than the whole cell. On the other hand, it is true that the resolution of descSPIM is not superior in comparison to other advanced microscopes (for example, Benchtop mesoSPIM with a lateral resolution of 1.5 $\mu\text{m} \times 1.5 \mu\text{m}$ [Vladimirov et al. 2023], or OTLS NODO with a lateral resolution of 0.45 $\mu\text{m} \times 0.45 \mu\text{m}$ [Glaser et al. 2022]). As the reviewer's recommendation, we have amended our terminology related to this point to more moderate expressions, such as “cellular resolution or more (P 14 L 326)” and “cellular level or more (P 18 L 413)”.

As the reviewer also pointed out, we do not necessarily advocate for the complication through upgrading the system with add-ons. In the revised manuscript, descriptions about technological enlargement with add-ons were omitted to avoid making positioning of descSPIM ambiguous. Also, we further emphasized positioning of descSPIM by citing and comparing the latest Benchtop mesoSPIM in *Discussions*.

Discussions (P 18, L 427-431) now reads:

While a recent Benchtop mesoSPIM was proposed to lower the cost of implementation, its construction is still difficult for biologists. In contrast to mesoSPIM/Benchtop mesoSPIM, the positioning of descSPIM is aimed to allow the end-users to utilize a personalized LSM, next to their bench used for tissue clearing.

Furthermore, to reinforce this point and in response to the recommendations of reviewers #1 and #3, we have included examples of community-driven system utilization in the revised manuscript (new **Fig. 5**). We believe that these modifications successfully clarify the system's positioning as a unique niche in the simple and affordable landscape.

Comment #2

The choice of non-motorized x, y and O stages makes precise tiling for large samples a challenge not only during acquisition but most importantly for image registration, tiling and fusion. In my opinion, image processing is a bigger challenge for end users than optics or cost which render the system not as practical. A motorized stage and an operating software such as micro-manager will allow to write the required metadata in the TIFFs file making the image processing a lot easier for non-experts. This choice also begs the question of how reproducible image acquisition can be? The manual rotation of a sample in a cuvette for instance can end in sample displacement making registration challenging. Could the authors, if they don't have quantified data on reproducibility, at least discussed this limitation in the discussion section?

Author's response:

We appreciate the insightful feedback concerning the challenges faced by end-users and the importance of reproducibility. As highlighted by the reviewer, we acknowledge that image processing can pose a burden. In response, we have shared semi-automated code on GitHub for registering $0^\circ/180^\circ$ images, particularly relevant when samples were rotated manually, as described in the original manuscript. We have additionally clarified this point in the revised manuscript to underscore the availability of our custom codes in the *Results* session.

Results (P 13, L 307- L 309):

In general, image processing poses a significant challenge for many end users in practical applications. To alleviate this burden, we have developed semi-automated custom codes tailored for registration and merging tasks.

Moreover, as highlighted, the reproducibility of image acquisition is indeed a critical issue. To ensure the reproducibility in a single image stack, we newly conducted multiple rounds of imaging for both manual initiation and saving of images, as well as using μ Manager-based custom operation software. This acquisition method was verified to enable reproducible image acquisition. Indeed, it effectively reduced axial displacement, which was observed in initial acquisition method. Please find the relevant *Supplementary Figs. 9-11*, and the added paragraphs as follows.

Results (P 12, L 262-271) now reads:

We validated the reproducibility of z -stack measurements and identified axial shifts in the stack resulting from disparities in start timing (**Supplementary Fig. 9a**). However, these discrepancies were effectively mitigated through adjustments solely in the z -direction (**Supplementary Fig. 9b**). Notably, no shifts were detected in the xy -plane, ensuring consistent acquisition reproducibility (**Supplementary Fig. 9a**). To address axial displacements, we provided custom software allowing users to specify start timing and ensure z -position (**Supplementary Fig. 10**). Similar to **Supplementary Fig. 9**, this acquisition method was verified to enable reproducible image acquisition (**Supplementary Fig. 11**). Indeed, it effectively reduced axial displacement, rendering it an ideal option for experiments requiring precise position reproducibility (**Supplementary Fig. 11**).

Discussions (P 19-20, L 455-463) now reads:

For controlling this minimal LSFM, we initially proposed a method of simultaneously driving two different device-associated software. While this procedure enabled reproducible image acquisition, axial displacement for a few slides might occur. The required manual synchronization made it difficult to achieve complete elimination of human error, and in this case, post-correction of displacement of a few slides is easy. To automate the z -stack acquisition, a low-cost USB to IO expansion board was employed to synchronize camera initiation with stage movement. The latter, the custom software-based approach excels in reproducibility, but due to the setup complexity, we recommend prioritizing the utilization of the former, device-associated software for its simplicity.

Materials and Methods (P 26, L 627-631) now reads:

In addition, a low-cost USB to IO expansion board (Arduino Uno Rev3) was employed to synchronize camera initiation with stage movement to automate z -stack acquisition (**Supplementary Fig. 10**). A simple Jupyter Notebook was developed to guide users through a z -stack acquisition (available on GitHub). The camera was controlled in μ Manager, and the stages were controlled with a napari-micromanager widget.

Regarding the reproducibility when using the manual θ stage, the precision largely relies on the competence of ANTs software. This registration tool has been used in multiple previous studies [Mano et al., 2021; Goubran et al., 2019; Johnson et al., 2023], supporting its reliability in practical use. However, as the reviewer pointed

out, there would be a potential issue in an inaccurate registration by variation in rotation angle between trials. We clarified this possible issue in the limitation part of the *Discussions* session.

Discussions (P 20 L 467-472) now reads:

We also utilized a manual θ stage along with single-side illumination and detection paths to enable imaging from various angles. This necessitated registration procedures to merge the acquired stacks. The ANTS software, which we employed for this procedure, has demonstrated reliability in several studies (59–61). However, it is worth noting that the manual rotation implemented in the current descSPIM setup may introduce some limitations in registration accuracy.

Comment #3

The choice of having a single illumination side is understandable to limit cost but having to take images at 2 angles can significantly increase the file size which now place the financial burden in data storage and processing. Could the authors provide and discussed data size for their examples?

Author's response:

Thank you for addressing this critical aspect of the system's practical use. As the reviewer highlighted, conducting two-directional imaging from 0° and 180° doubled the data size. For instance, in the case of the Thy1-YFP-H whole mouse brain (Figure 3), each dataset comprised 997 images in 12-bit TIFF format ($3.45 \times 3.45 \times 10 \mu\text{m}^3$, 2160×4096 pixels), resulting in a data size of 8.8 gigabytes per dataset. Consequently, the total data size across the entire brain (eight stacks) reached 70 GB. Following the registration and reconstruction of these eight tiles into a single whole-brain image, the final data size in TIFF format was 22.4 GB. This data size is indeed comparable to our advanced LSFM system (GEMINI): in this case, the 16-bit TIFF stack ($6.5 \times 6.5 \times 6.5 \mu\text{m}^3$, 2160×2560 pixels) covering a single whole mouse brain (see Supplementary Fig. 16) amounted to 14 GB. Similar to GEMINI, the mesoSPIM utilized a 2048×2048 pixel sCMOS camera for image acquisition with $6.5 \mu\text{m}$ iso-voxels, resulting in a potential data size of 12-16GB for the entire mouse brain. Therefore, all of these systems yield image data within the range of tens of gigabytes. From the perspective of operational and storage costs, there is no significant difference. Hence, we believe that our application involving multiple directional imaging does not impose a burden when users employ adequate workstation and storage equipment for general volume data processing and analysis. We have clarified these points in our revised manuscript.

Results (P 14-15, L 335-338) now reads:

Moreover, since the final data size is approximately the same for both the Gemini and multi-view datasets, we believe that handling data from multiple stacks obtained through this acquisition method would not pose a significant burden on modern 3D imaging systems that handle recent data.

Materials and Methods (P 33, L 785-788) now reads:

The raw images of each tile were composed of TIFF stacks with a lateral resolution of 3.45 μm , an axial resolution of 10 mm, and consisted of 997 slices. The data size of each tile was 8.8 GB. The whole mouse brain image was reconstructed from these eight tiles as described, resulting in a final data size of 22.4 GB.

Comment #4

The comparison between CUBIC and BABB treated samples is important and unfortunately only purely qualitative. Could the authors provide at least PSF measurements and light sheet dispersal quantification in various RIMS broadly used? (Maybe adding Ethyl Cinnamate?).

Author's response:

As pointed out by the reviewer, it is crucial to measure the point spread function (PSF) in organic solvents commonly used for tissue clearing, such as ethyl cinnamate and BABB, and compare it to that in CUBIC. However, it is worth noting that common fluorescent beads coated or combined with dyes can become unstable or dissolve in organic solvents. Indeed, widely used fluorescent beads, including Siacstar-redF (#40-00-103, micromod Partikeltechnologie GmbH) and Nile Red, were unable to be used for quantifying the PSF in organic solvents, at least by our hand. To fully address the reviewer's concern, we widely searched for fluorescent beads that are compatible with these organic solvents and finally found out several quantum dot products (Qdot 605 ITK amino (PEG) quantum dots, #Q21501MP, Invitrogen) were utilized for this purpose. The corresponding data has been included in **Supplementary Fig. 12** and the subsequent sections. Furthermore, we would like to underscore the importance of our discovery regarding the suitability of Qdot for use in organic solvents. This discovery is crucial for the microscopy field with tissue clearing, as previous studies have not documented successful PSF measurements in such organic solvents.

Results (P 12-13, L 279-293) now reads:

To evaluate PSFs in BABB (2) and ECi (3), we estimated their lateral and axial resolutions in FA mode by FWHM values of the intensity profile of the signals from a $\Phi 15$ nm quantum dots embedded in gelatin gel substituted with organic solvents (**Supplementary Fig. 12a**). The lateral (x - y) resolutions were measured as 4.8 ± 0.3 μm in BABB and 4.6 ± 0.4 μm in ECi, showing negligible deviation from the values obtained in CUBIC-R-agarose (**Fig. 1g**). However, the axial resolutions (x - z or y - z) were found to be 9.5 ± 0.7 μm in BABB and 9.8 ± 0.3 μm in ECi, representing a degradation of over 30% compared to the values in CUBIC-R-agarose (**Fig. 1g**). This degradation stemmed not only from the longer excitation wavelength of approximately 50 nm but also from spherical aberration resulting from the RI difference between the organic solvent and the glass cuvette (RI = 1.56 for BABB and ECi and 1.52 for BK7 glass). By extrapolation of FWHM values to the literature-defined effective NA value of cylindrical lenses for SPIM (18) (**Supplementary Fig. 4b**), we determined the effective NA values for the FA mode to be 0.026 in BABB and 0.025 in ECi. Utilizing these NA values, the eFOV in FA mode was estimated to be approximately 1488 μm in BABB and 1600 μm in ECi.

Materials and Methods (P 27-28, L 651-661) now reads:

Furthermore, we assessed the point spread functions (PSFs) in FA mode using organic solvent-based clearing reagents, specifically BABB and ECI. Quantum dots (Qdot™ 605 ITC™ Amino (PEG) Quantum Dots; Thermo Fisher Scientific) were diluted (1:100, v/v) in 2% agarose (w/v) + 2% gelatin (w/v, #16631-05, Nacalai) gel. The gel underwent dehydration with ethanol (50% for two hours, 70% for two hours, 100% for two hours, 100% overnight), followed by refractive index adjustment using either BABB or ECI as the organic solvent. Subsequently, the gel containing the quantum dots was placed into a cuvette for measurement. The fluorescent images of the quantum dots had a pixel size of 3.45 μm . We acquired 150- μm -thick z -stacks at 3 μm intervals and used them to reconstruct xyz images of the beads. To visualize the fluorescent signals from the quantum dots, the 561 nm excitation laser light (7 mW output) and the 580–620 nm bandpass filter were selected for measurements.

Comment #5

In most of the qualitative figures, such as Fig3, the authors only zoom in on the side of the tissue where the illumination and detection will be optimal. Can they provide images of the inside of the tissue to visualize the loss of resolution across?

Author's response:

Thank you for the reviewer's comment. We newly added *Supplementary Fig. 12* to provide a series of xy -plane images covering the entire brain of PI-stained Thy1-YFP-H mouse. This data demonstrated that the internal structure of the specimen was adequately imaged.

Comment #6

L227: "overall quality was comparable to our advanced light-sheet microscopy system". This is again, a stretch. If no quantification, could the authors provide side by side image comparisons of the 2 system at the same magnification in supp Fig11?

Author's response:

Thank you for the reviewer's comment related to the previous issue. We have updated our manuscript to include quantitative comparison of the resolutions of these two microscopy systems. For this purpose, we obtained the intensity profiles of somas in images of PI-stained Thy1-YFP-H mice. As a result, we found that the axial resolution of descSPIM in FF mode was $32.0 \pm 1.1 \mu\text{m}$ ($n = 20$), while the GEMINI's resolution was $17.5 \pm 1.6 \mu\text{m}$ ($n = 20$). We have incorporated these quantitative results into the findings and revised the wording from "comparable" to "did not reach the similar level." We believe that this change addresses the ambiguity associated with the qualitative term "comparable" and provides a more quantitative and equitable basis for comparison.

Results (P 14-15, L 324-338) now reads:

The resulting two-color image of the entire brain revealed the distributions of cell bodies and neurites, indicating that descSPIM provided 3D neural images with cellular resolution or more (**Fig. 3b**). For further

comparison, we acquired a 3D image of another *Thy1*-YFP-H brain with our advanced light-sheet microscopy system (GEMINI (39)). Comparing the FWHM of the axial intensity profile of somas imaged by descSPIM (Fig. 3b) and GEMINI (Supplementary Fig. 16), we found that the axial resolution of descSPIM in FF mode was $32.0 \pm 1.1 \mu\text{m}$ ($n = 20$), while the GEMINI's resolution was $17.5 \pm 1.6 \mu\text{m}$ ($n = 20$). Therefore, although descSPIM can support the 3D imaging applications on an organ-scale required for neuroscience research, its axial resolution did not reach the similar level when compared with an advanced system. Switching to high-resolution FA mode would be necessary in some applications requiring a higher axial resolution.

Comment #7

The usefulness of the descSPIM for pathologist should be either removed or at least toned down. As the authors state, they obtained “relatively low resolution for usual pathology examination”. L274 should be removed since it is not a scientific argument “a pathologist concurred...”.

Author's response:

Thank you for the reviewer's comment. According to your suggestion, we removed the original description of the utility of descSPIM in pathology and revised the statement as follows.

Results (P 16, L 382-384) now reads:

Nevertheless, for future diagnostic objectives, descSPIM has the potential to reach not only the basic sciences but also the clinics.

Comment #8

Details and typo/phrasing:

1. L61: I think the authors mean “adapted” instead of “adopted”
2. Replace several instances of “obtained” with something like “generated” for data they produced (l247 and 267).
3. SuppFig1: on the mesoSPIM line I think the authors mean “Facility” instead of “Faculty”.

Author's response:

Thank you for your detailed comments regarding typographical errors and misphrasing. We have corrected each section as you pointed out.

Introduction (P 6, L 125-126) now reads:

Such low NA systems have been improved in both image quality and simplicity by adding various options adapted to the specimen (37–40).

Results (P 15, L 355-358) now reads:

We generated a 3D image of the entire tumor mass of a cancer cell line-derived xenograft (CDX) of BT-474 human breast cancer cell line, labeled with CD31-FITC (vessel marker) and PI, for application in drug discovery.

Results (P 16, L 375-376) now reads:

we also generated a 3D image of a 2 mm-thick CDX section with FA mode and TLS.

In **Supplementary Fig. 1**, the ‘faculty’ has been changed to ‘facility’:

	Schematics	Main objects	Control	Components	Target to be placed	Price	Reference
openSPIM	SPIM *L-: conventional; X-: with dual-angle illumination and detection	Living small samples	Micromanager-based	Custom-made parts included	Laboratory	21k USD- (L-) 35k USD- (X-) single ex. wavelength and sCMOS camera excluded	Pitrone P. G., et al., Nat. Methods (2013); Girstmair J. et al., Adv. Biol. (2021).
OpenSpin Microscopy	SPIM, DSLM and OPT	Living small samples and meso-scaled samples	Micromanager-based	Custom-made parts included	Laboratory or Facility	NaN	Gualda E. J. et al., Nat. Methods (2013).
mesoSPIM	SPIM with axial sweeping and dual angle illumination and detection	Meso-scaled cleared tissues and animals	Python and PyQt5-based	Custom-made parts included	Facility	162k USD version 5 with 4-ex. wavelength	Voigt F. F., et al., Nat. Methods (2019); Valdimirov N., Voigt F. F., et al., bioRxiv (2023)
Benchtop mesoSPIM	SPIM with axial sweeping and dual angle illumination and detection	Meso-scaled cleared tissues and animals	Python and PyQt5-based	Custom-made parts included	Facility	95k USD with 3-ex. wavelength	Valdimirov N., Voigt F. F., et al., bioRxiv (2023)
descSPIM	SPIM	Meso-scaled cleared tissues	Devise-associated software	All commercially available parts	Laboratory or personal	20k-50k USD depending on the number of ex. wavelength (1-4)	This paper

Dear Reviewer #3

We fully appreciate your thoughtful and encouraging comments. Here, we reply to all the concerns raised by the reviewer #3. We believe that this revision improved the quality of our manuscript significantly. Here are our responses to your insightful comments.

Comment #1

The descSPIM is presented as an open science tool and it is disappointing to see that the community aspect is in the discussion. This is a result as it is rare to get a new tool which as already been used and develop outside its original lab, with several workshops. It will be important to relocate this element in the result section with maybe a figure showing a map of the already expanding new descSPIMs are, and pictures from the workshop. It is essential that new tools are not single machines in a lab but already replicated before to be published and this is there so please show it.

Author's response:

We extend our appreciation for your thoughtful and constructive feedback. In response to Reviewer #1's third comment, we acknowledged that the widespread implementation of our system across numerous facilities is a significant outcome. This information has been included as **Fig. 5** in the revised manuscript. **Fig. 5a** presents a map of descSPIM installations, along with accompanying photographs of the descSPIM equipment at each facility. **Fig. 5b-i** display images captured using descSPIMs specific to those locations. We believe this clarification underscores the community-oriented nature of descSPIM, highlighting its substantial reproducibility and broad applicability across various biomedical domains.

Comment #2

the introduction of LSFm is a bit simplistic and only discussed the SPIM and the DSLM and nothing on the single objective, the mirror based approach and so on so it will benefit from a better presentation as it is currently confusing as diSPIM is presented much later. The DSLM is not even properly referenced!!!

Author's response:

We fully agree with the reviewer's suggestion that further elaboration on LSFM would enhance the readers' understanding. In addition to SPIM and DSLM, we have included descriptions of implementations such as tiling LSFM, non-diffracting beam-based light-sheets, and optical lattice-based light sheets. Furthermore, we have added a consolidated explanation of dual-view plane illumination SPIM (diSPIM), open-top light-sheet (OTLS) microscopy, swept confocally-aligned planar excitation (SCAPE), and single-objective SPIM (soSPIM) in **Introduction**.

Introduction (P 5-6, L 95-120) now reads:

Two significant concepts of LSFM, selective plane illumination microscopy (SPIM) and digital scanning light-sheet microscopy (DSLIM), were proposed prior to the extensive development of tissue clearing techniques in 2010's (16, 17). Their original purpose was to image small and intrinsically translucent specimens such as fly embryo and fish larva in 3D and 4D manner (16, 17). SPIM utilizes a cylindrical lens to generate a static excitation light sheet. DSLIM generates a light-sheet by scanning a narrow-focused beam in one axis using a galvanometer mirror. Most excitation light-sheets utilized in both SPIM and DSLIM are generated by focusing a Gaussian beam, causing a tradeoff problem between axial spatial resolution and effective field of view (eFOV) (18). To overcome this, tilling LSFM (19) and axial sweep schematics (20) were proposed and successfully enlarged eFOV with maintaining axial spatial resolution. DSLIM has allowed for more advanced approaches than SPIM, instead of a bit more complex system (18). One of the most prominent examples is noise reduction by synchronizing the galvo-mirror with the rolling shutter of the camera (21, 22), which has a similar effect to the confocal slit detection (23). Moreover, recent state-of-the-art approaches utilizing non-diffracting beam-based light-sheets such as Bessel beam-based (24–26), Airy beam-based (27) and optical lattice-based light sheets (28) are reported based on the DSLIM implementations.

Most LSFMs have adopted an orthogonal configuration between the excitation light path and detection light path, inducing physical restrictions to place specimens. Thus, general holders, such as culturing dishes and well-plates, were difficult to be applied. Tilting illumination-detection configuration-based inverted geometry systems, such as dual-view plane illumination SPIM (diSPIM) (29, 30) and open-top light-sheet (OTLS) microscopy (31–33), have been developed to provide ease of use in measurements of wider samples. Moreover, single-objective lens-based LSFMs, such as swept confocally-aligned planar excitation (SCAPE) (34, 35) and single-objective SPIM (soSPIM) (36) have also been implemented for achieving high-spatial-resolution imaging.

Comment #3

the use of "basic" in the results does not make sense as the only advanced part concerned processing and not the system itself. Moreover Figure 1 is nice but need some works as first the panel a does not defined what "adanaced" means to the authors? the cost could be nummbered as the price of the descSPIM is provided as well as the 2 mesoSPIM options and it will be good to also mention the desktop version of the mesoSPIM which is already published rather than focused on the most expensive one!

Panel b is fine but the colour coding is not reuse in the next panels. Panel c and d showing perspectives of a blakc block without much shadow is not helping. Maybe a better choice of colour and shading including transparency should improve the reader read-out of the scope structure and please check your spelling: Excitation not Excitaion (panel c). And using the exact same angle of perspective without overlappng labels on the design will help too! Then in panels e to g this is an overwhelming amount of data per inch square. Why do you need to spell out the FWHM already explained in the text and visibe in the panel above, you just repeat the information and the figure will be so small that it makes no sense, please do improve this!

Author's response:

We appreciate the reviewer's comprehensive remarks. As this comment contains multiple elements that require modification, we elaborate on our responses for each as follows:

About the use of "basic" :

We agreed that "basic" qualifying descSPIM in the results does not make sense, therefore we removed "basic" from the titles of *Results* in "P 7 L 164" and "P 11 L 242",

About what "adanaced" means :

We have added a definition of what we considered as 'advanced' to the introduction.

Introduction (P 6, L 136) now reads:

On the other hand, the cutting-edge custom-built microscopy, plotted as the advanced system in **Fig. 1a**,

About the 2 mesoSPIM options :

We fully agree with the reviewer. It is crucial that efforts are ongoing to reduce the cost of benchtop mesoSPIM, and we have added a plot for the benchtop mesoSPIM in panel *Fig. 1a*. and also in *Supplementary Fig. 1*.

About *Fig. 1b* :

As you mentioned, we indeed assigned magenta for FF mode and orange for FA mode. However, we inadvertently omitted this rule in *Fig. 1b* of the previous manuscript. We have rectified this oversight in the current version.

About *Fig. 1c and d* :

Thank you for your insightful and detailed suggestion on how to illustrate our system's configuration to make it more comprehensible to readers. We have added new schematics of shaded and wireframe models from different angles as *Supplementary Fig. 2*.

Also, we appreciate that you pointed out the typographical errors in Fig. 1c. We have made the necessary corrections.

About an overwhelming amount of data:

Thank you for bringing that to our attention. It's true that the previous version contained excessive information. In this revised version, we have omitted the details from the representative intensity profiles. However, we retained the textual description of the mean and standard deviation (SD), and number (N). This decision was made as we introduced a new *supplementary Fig. 12*, according to Reviewer #2's comment #4, which includes for comparison.

Comment #4

acronym bonanza: you use an acronym in the abstract but it is undefined and along the text many acronyms paved the way to SAOD (severe acronyms disorders). It is hard in those 2 fields to avoid using them but then use them and define them as CUBIC and others are just referenced so many could get the same faith!

Author's response:

We appreciate your input regarding the overuse of acronyms. We acknowledge that they did, in fact, reduce the legibility. We ensured the accurate spelling of each term and incorporated citations where appropriate in the revised manuscript. We sincerely hope that this amendment will assist in mitigating your SAOD symptom.

Introduction (P 4-5, L 80-94) now reads:

Since the early attempts of organ-scale three-dimensional (3D) imaging with light-sheet fluorescence microscopy (LSFM) (1), tissue clearing has become the gold standard for volumetric tissue, organ, and body imaging. Highly efficient tissue clearing techniques for LSFM imaging, such as benzoic acid benzyl benzoate (BABB) based clearing methodology (2), ethyl cinnamate (ECi) based clearing methodology (3), 3D imaging of solvent cleared organs (3DISCO) (4), clear, unobstructed brain/body imaging cocktails and computational analysis (CUBIC) (5, 6), *m*-xylylenediamine (MXDA)-based aqueous clearing system (MACS) (7), polyethylene glycol (PEG) associated solvent system (PEGASOS) (8), small micelle improved human organ antibody efficient labeling (SHANEL) (9), cleared lipid extracted acryl hybridized rigid immunostaining/in situ hybridization compatible tissue hydrogel (CLARITY) (10), and stabilization under harsh conditions via intramolecular epoxide linkages to prevent degradation (SHIELD) (11), have provided detailed protocols or have already been commercialized, enabling end-users to easily approach these techniques. The maturation of this field has supported numerous scientific discoveries in the wide range of biomedical research (12–15).

Comment #5

title: it seems that using microsocpy is not correct as you are presenting not a technique but an instrument so it should be microscope...

Author's response:

Thank you for your suggestions concerning the title from a semantic viewpoint. Reviewer #1 also kindly recommended revising the title, which has been updated as follows.

Title:

descSPIM: an affordable and easy-to-build light-sheet microscope optimized for tissue clearing techniques

Comment #6

In results section 1 you mention device associated software? COuld not find any mention of it in the method? As well later you state minute-order volumetric imaging per stack, any value?

Author's response:

Thank you for the comments regarding the description of the software. The information has been in the **Materials and Methods** section. Furthermore, based on the suggestion from Reviewer #1 to use software like μ Manager for acquisition instead of our original manual procedures, we have also included a description of the method for capturing images without using the software provided with the camera and stage.

Materials and Methods (P 26, L 610-631) now reads:

When obtaining a z-stack image of a mouse hemisphere with FF mode (**Fig. 2a**), the velocity of the sample stage (v_{stage}) was set to 50 $\mu\text{m}/\text{sec}$ and the velocity of the detection optics (v_{detect}) to 17.3 $\mu\text{m}/\text{sec}$ on the actuator-associated software KINESISTM (Thorlabs). The traveling distance of the sample stage was 10 mm

($d_{z_{\text{stage}}} = 10 \text{ mm}$, $d_{z_{\text{sample}}} = 3.46 \text{ mm}$). The time-lapse image was obtained with the exposure time of 200 ms. The resulting xyt image was converted to xyz format with a z -range of 10 mm and a z -interval ($= v_{\text{stage}} \times \text{exposure time}$) of 10 μm using ImageJ/Fiji (<http://fiji.sc/Fiji>) (64). Due to the specification of the camera-associated software ThorCam™ (Thorlabs), files were split when they exceed 1 GB. Therefore, a final stack was generated using ImageJ/Fiji's concatenate function. As for a mouse coronal section in FA mode (**Fig. 2b**), v_{stage} was set to 25 $\mu\text{m}/\text{sec}$ and v_{detect} to 8.5 $\mu\text{m}/\text{sec}$. The traveling distance of the sample stage was 10 mm ($d_{z_{\text{stage}}} = 10 \text{ mm}$, $d_{z_{\text{sample}}} = 3.40 \text{ mm}$). The time-lapse images of multiple stacks for TLS were obtained with the exposure time of 200 ms. The resulting xyz images had a z -range of 10 mm and a z -interval of 5 μm . To visualize the fluorescent signals from PI, the 515 nm excitation laser light (3 mW output) and the 550 nm longpass filter were selected for measurements. We used a notebook PC (Intel® Core™ i7-8750H, 16GB RAM, SSD 256GB M.2 2280 S3-M SDAPNUW-256G) for the operation and the initial data processing.

In addition, a low-cost USB to IO expansion board (Arduino Uno Rev3) was employed to synchronize camera initiation with stage movement to automate z -stack acquisition (**Supplementary Fig. 10**). A simple Jupyter Notebook was developed to guide users through a z -stack acquisition (available on GitHub). The camera was controlled in $\mu\text{Manager}$, and the stages were controlled with a napari-micromanager widget.

We intended to convey that 3D imaging of the target sample without slicing is possible within tens of minutes. Therefore, the sentence has been changed as follows.

Result (P 11, L 238-239) now reads:

This method enabled rapid volumetric imaging, typically finishing an acquisition of a single z -stack in minutes.

Comment #7

lines 216 to 218: you explain about fusion issues and say they are "globally" rather vague, isn't it. It is probably due to the limited number of views you use and the limited overlap as well as the detection decay of the signal.

Author's response:

Thank you for the reviewer's comment. We admit that the definition of "global" and "local" registration was somewhat a vague expression in the previous manuscript. The registration process described herein consists of two primary steps (detailed in **Materials and Methods, Supplementary Figs 13, 14**). Initially, we conduct a general registration of the entire field using images obtained from the nuclear staining channel, employing linear transformations exclusively. This approach is chosen because the signal from the target channel (e.g., YFP in the Thy1-YFP mouse model) may not uniformly permeate the entire observation area. By establishing a broad framework for registration during this initial stage, our aim is to enhance overall accuracy. Subsequently, we refine the registration process for individual signals using images from the target channel, incorporating non-linear transformations as necessary. In previous iterations of this manuscript, the initial step was labeled as 'global' and the subsequent step as 'local'. However, this terminology could potentially imply variability in the scope of the registration calculation itself, which may lead to confusion. In response to your comment, we have

revised 'global' to 'brain-wide' and 'local' to 'pixel order'. We believe that this revision adequately clarifies the registration process we carried out and the meaning of "local" and "global" registration regarding this procedure.

Results (P 14, L 313-316) now reads:

The brain-wide registrations seemed to be completed in both the 25% and 50% downsized (*Supplementary Fig. 14b*). When examining the images in pixel order, however, misaligned signals were observed when using 25% compressed data:

Comment #8

equations lines 466 and 468.. it should be spelled dZsample and not dZsapmle

Author's response:

Thank you for your comments. We have corrected the typographical errors in *Materials and Methods* (P 25, L 603).

Comment #9

line 525 please show soem respect to other open science project and please reference Fiji not only a mention, not cool!

Author's response:

Thank you for reminding me of the necessity to properly cite Fiji's original paper; I have included the citation in *Materials and Methods* (P 26, L 616).

Closing

I do hope this long list of revisions is not scaring the authors and the editors as this paper should get published as not only users needs it but also maybe developing countries where LSFMF is unfortunately a luxury.

Author's response:

The mass number of the reviewer's insightful critiques deeply frightened us at first. However, we now believe that our revised manuscript has fully addressed all of the concerns and has significantly improved over the original version. We again appreciate your dedication to emphasizing the significance of our manuscript.

Reviewers' Comments:

Reviewer #1:

Remarks to the Author:

The authors have satisfactorily responded to all my comments and revised the manuscript accordingly.

Reviewer #2:

Remarks to the Author:

The authors convincingly addressed all my suggestions by either amending the text or generating additional experiment. It will be a very useful and important new system that will reach a large population throughout the globe.

Congratulations!!

Reviewer #3:

Remarks to the Author:

The DecSPIM paper has been revised and the new version is much better and will be able to convince many users to build their own systems to test new clearing protocols and screen samples.

This step will allow them to improve their work and time on the microscope.

I am glad to have been part of this venture as a reviewer and I am looking forward to encountering many systems here and there.

Dear Reviewers

Thank you for your comments on our manuscript: descSPIM: an affordable and easy-to-build light-sheet microscope optimized for tissue clearing techniques. We are pleased to note favorable comments from the reviewers.

Dear Reviewer #1

Comment

The authors have satisfactorily responded to all my comments and revised the manuscript accordingly.

Authors' response:

We would like to thank the reviewer #1 for the important comments, which enabled us to improve the quality of our manuscript.

Dear Reviewer #2

Comment

The authors convincingly addressed all my suggestions by either amending the text or generating additional experiment. It will be a very useful and important new system that will reach a large population throughout the globe.

Congratulations!!

Authors' response:

We appreciate the reviewer #2 for indicating solid points, importantly reaffirmed where our system stood.

Dear Reviewer #3

Comment

The DecSPIM paper has been revised and the new version is much better and will be able to convince many users to build their own systems to test new clearing protocols and screen samples.

This step will allow them to improve their work and time on the microscope.

I am glad to have been part of this venture as a reviewer and I am looking forward to encountering many systems here and there.

Authors' response:

We truly thank reviewer #3's encouraging suggestions to improve our manuscript for unparalleled one.